



# A correction method for large deflections of cantilever beams with modal approach

Ozan Gözcü[1], Emre Barlas[2], and Suguang Dou[1]

[1]DTU Wind Energy, Technical University of Denmark (DTU), Frederiksborgvej 399, 4000 Roskilde, Denmark
[2]Ørsted, Nesa Allè 1, 2820 Gentofte

**Correspondence:** Ozan Gözcü(ozgo@dtu.dk)

**Abstract.** Modal based reduced order models are preferred for modelling structures in engineering problems due to their computational efficiency. One of the important limitations of the classic modal based models is that they are linear and thus can not fully account for the nonlinearities in structures. This study proposes a fast correction method to account for geometric nonlinearities linked to large deflections in cantilever beam-like engineering structures. The large deflections cause secondary

motions such as axial and torsional motions when the structures go through bending deflections. The method relies on pre-computed correction terms and thus adds negligibly small extra computational cost to the time domain analyses of the dynamic response. The accuracy of the method is examined on a straight beam benchmark model and an engineering wind blade model for the IEA 15 MW wind turbine. The results show that the proposed method increases the accuracy of modal approach significantly in estimating the secondary motions in comparison to the conventional modal based linear models.

**Keywords** : Geometric nonlinearity, modal reduction, modal derivatives, expansion modes, IEA 15MW turbine

## 1   Introduction

Reduced order models (ROMs) based on modal approach is used in many structural engineering problems such as wind turbine blades (Hansen, 2015), aircraft wings (Bisplinghoff et al., 2013), space crafts (Marshall and Pellegrino, 2021) due to their computational efficiency and reasonable accuracy. Most of the existing ROMs are based on the small deflection assumption,

in other words the stiffness, mass and damping matrices are not updated by deflections. Hence, the accuracy of modal based ROMs reduces as deflections increase and their errors become significant for applications with large deflections such as long and flexible wind turbine blades or aircraft wings. Moreover, the error in structural response amplifies the error in aeroelastic response and load analysis due to the coupled nature of problem.

The large deflection effects on aeroelastic stability and loads for wind turbine blades (Kallesøe, 2011; Beardsell et al., 2016;

Collier and Sanz, 2016; Riziotis et al., 2008) and aircraft wings (Cesnik et al., 2014) are now well known. Although these effects can be modeled in some aeroelastic tools by using geometrically nonlinear structural solvers (Larsen and Hansen, 2019; DNV, 2016; Wang et al., 2017; Bauchau, 2009), linear modal based ROMs are still in use even for structures with large deflections such as wind turbine blades (Øye, 1996; Jonkman and Buhl Jr, 2005) due to their speed.



The focus of this study is cantilever beam structures and their reduced order models (ROMs) based on modal approach used in coupled simulations such as aeroelasticity and load simulation of wind turbines and aircraft. A new correction method for capturing geometric nonlinear effects in these ROMs is proposed. The method has possible minimum computational cost during the coupled analysis since it includes only some correction terms which don't require any extra iteration during the response analysis.

There are many studies in the literature on geometrically nonlinear ROMs and most of them focus on clamped-clamped beams or simply supported panels. Table 1 shows some prominent works with cantilever structures from the literature. All of these studies, except Gözcü and Dou (2020) are limited to 2D beam models with forces applied in single direction or the deflections do not exceed $25\%$ of beam span. Moreover, all the existing studies about geometrically nonlinear ROMs uses nonlinear stiffness terms which require iterations and slow down the computation of response analysis. The proposed method may be less accurate compared to the ROMs with nonlinear stiffness terms, however it is very fast and easy to implement to existing aeroelastic tools compared to them.

**Table 1.** Overview of the studies with geometrically nonlinear ROMs for cantilever structure models. The methods, reduction bases and example structures used in the reference studies are given together with applied forces and maximum deflections in terms of span length.

| Reference | Method | Reduction basis | Example structure | Force direction | Max. deflection |
|---|---|---|---|---|---|
| Kim et al. (2009) | Displacement based Non-intrusive with von Karman kinematics | Bending modes + Dual modes | 2D straight beam | 1 bending direction | 50% of span |
| Wang et al. (2013) | Displacement based Non-intrusive with von Karman kinematics | Bending modes + Dual modes | An unmanned aircraft wing | 1 bending direction | 25% of span |
| Jain et al. (2017a) | Intrusive with von Karman kinematics | Quadratic basis & Modal derivatives | A wing model | 1 bending direction | 2% of span |
| Rutzmoser et al. (2017) | Intrusive with von Karman kinematics | Different quadratic bases | 2D straight beam | 1 bending direction | 30% of span |
| Wu et al. (2018) | Intrusive with von Karman kinematics | Rubin basis + Modal derivatives | NREL 5MW blade | 2 bending direction | 1% of span |
| Gözcü and Dou (2020) | Force based Non-intrusive | Linear modes + Modal derivatives | 3D straight beam NREL5MW blade | 2 bending directions | 20% of span |
| **This study** | Linear approach Correction terms | Linear modes | 3D straight beam IEA15MW blade | 2 bending directions | 20% of span |





All studies given in Table 1 include nonlinear stiffness terms in the calculation and use either intrusive approach (Jain et al., 2017b) or non-intrusive approach (Mignolet et al., 2013) to compute these terms. This study uses linear approach which is same as the existing ROM analysis and it needs only linear modes in its reduction basis unlike the other studies given in the table.

The paper is divided into four sections. The kinematics of cantilever beam problem relevant to this study is explained in Section 2 and the proposed method is explained in Section 3. Example cases are introduced and their results are given together with discussion in Section 4 and the conclusion takes place in Section 5.

## 2  Relevant Kinematics

This section explains the kinematics of cantilever beams, geometric nonlinearities effects for symmetric beams and initially
curved beams such as wind turbine blades.

Most of the research studies about geometrically nonlinear ROMs focus on clamped-clamped beams or simply-supported plates. In clamped-clamped case, the nonlinearity arises due to the length change where a lateral deflection actually alters the length of the structure which causes additional stiffness effect such as hardening or softening depending on the sign of length change. This effect is called as membrane effect or bending-extension coupling (Touzé et al., 2021). In case of cantilever beams,
lateral deflection due to lateral forces doesn't result in length change (no axial strain), so there is no bending-extension coupling for cantilever beams. However, the free end of a cantilever beam displaces in beam span direction to keep the length constant when it bends as shown in Figure 1. In other words, the cantilever beams with lateral loading go through large rotations which don't result in strain.

The large rotations of cantilever beams change the geometry of the structure. For instance, an initially symmetric beam
doesn't behave as one in its deflected state, so it can show bending-torsion coupling. Figure 2 depicts these kinematics using a

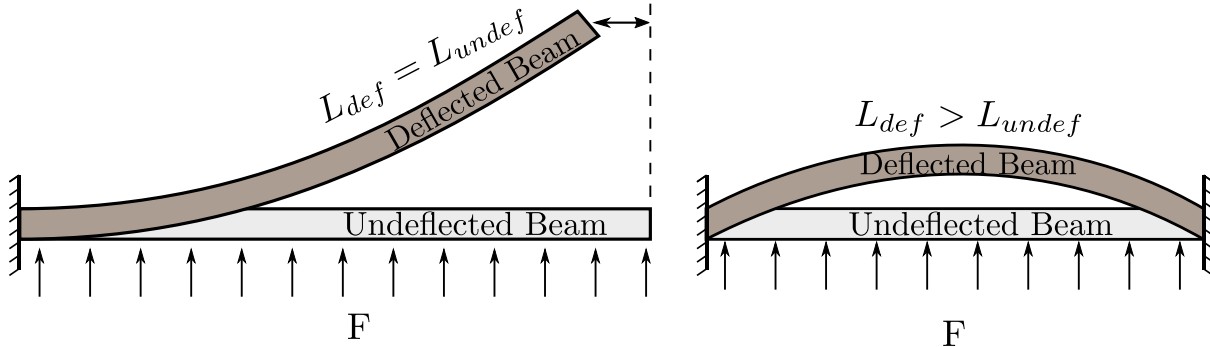

**Figure 1.** Displacements of a cantilever and a clamped-clamped beam under lateral distributed loads $F$. The deflected beam length ($L_{def}$) is equal to the undeflected length ($L_{undef}$) for the cantilever beam, resulting in zero axial strain. The clamped-clamped beam has axial strain due to elongation in beam length.





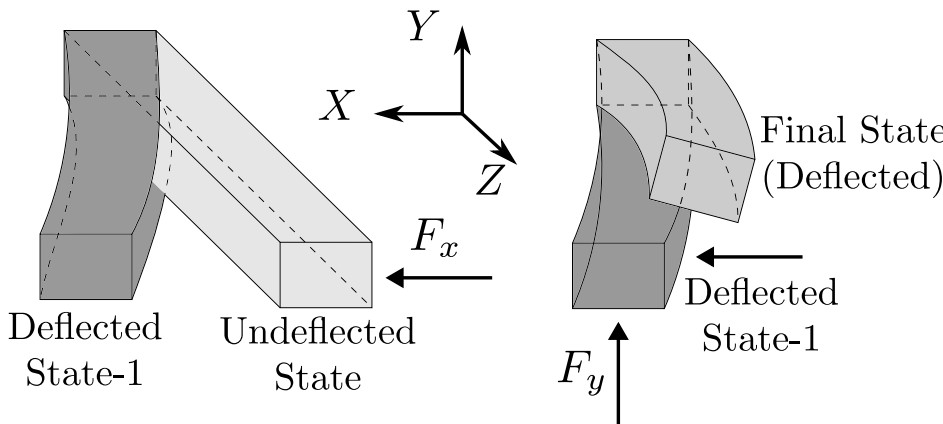

**Figure 2.** Illustration of beam deflections in lateral ($x-$, $y-$) directions and the resulting torsional motion due to the bending/torsion coupling at the deflected state. The coordinate system is given at beam root where no deflection occurs.

straight cantilever beam. The lateral force ($F_x$) deflects the beam to *State-1*. Subsequently, $F_y$ force which is perpendicular to $F_x$, is applied on the beam and beam reaches its *Final State*. It is observed that the beam motions occur not only in lateral ($x-$, $y-$) directions but also in torsion direction, even though no torsion load applied on the structure. Geometrically linear models (such as ROMs based on modal approach) fail to capture such kinematics.

These kinematics become even more prominent for applications such as wind turbine blades which are initially curved structures due to prebend, aerodynamic twist and back swept. They have couplings between bending and torsion motions at their undeflected states already due to their curved and twisted geometry but the lateral deflections change the magnitude as well as the direction of these couplings. Figure 3 shows the torsion deflection due to the combination of flapwise and edgewise deflections of a wind turbine blade with prebend at its deflected position. The edgewise and torsion motions are already coupled

for a blade with prebend at its undeflected state as shown by blue dashed lines in the figure, however this coupling first reduces and even change its sign by flapwise bending as shown by red continuous lines. This also alters aeroelastic stability and loads of the blades in an aeroelastic analysis (Kallesøe, 2011) and it becomes significant for very flexible wind turbine blades. In case geometrically linear models like classical ROMs based on modal reduction are used, the change of torsion-edgewise coupling cannot be captured.

## 3   Method

The nonlinear geometric effects on a cantilever beam can be captured by nonlinear ROMs using different methods (Intrusive or Non-intrusive see Table 1) and reduction basis where linear modes are used together with quadratic vectors such as `Expansion modes` (Hollkamp and Gordon, 2008) or `Modal derivatives` (Idelsohn and Cardona, 1985). It is observed that effect of nonlinear stiffness terms are not significant for moderately large deflections of a cantilever beam. On the

other hand, the geometric nonlinear effects on deflections explained in Section 2 are apparent for cantilever beams undergo-



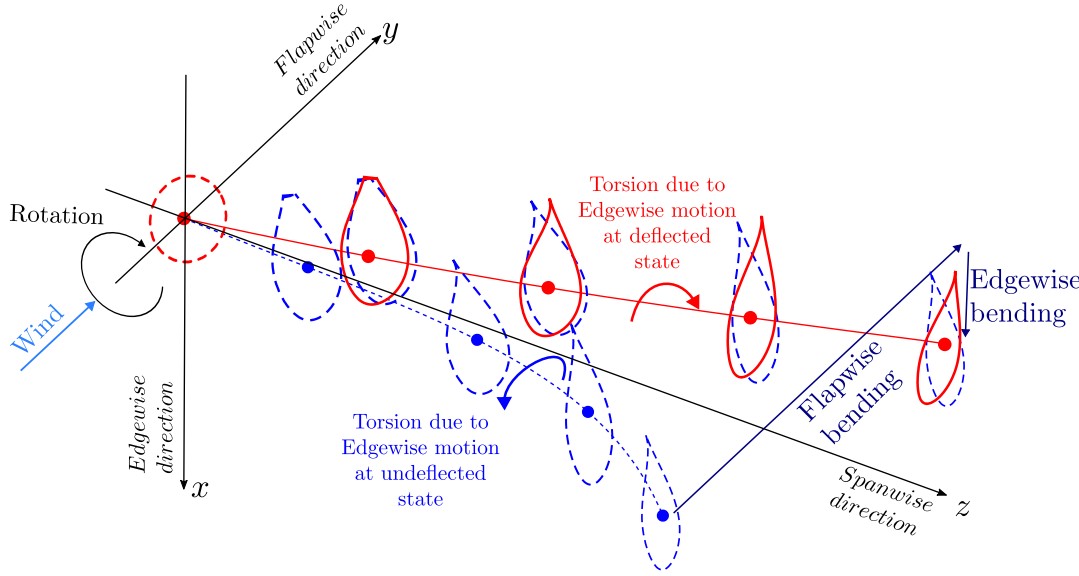

**Figure 3.** Illustration of a wind turbine deflections in flapwise ($y-$) and edgewise ($x-$) directions and the resulting torsional motion due to the bending/torsion coupling at the deflected state. The lateral ($x-$, $y-$) and axial ($z-$) directions are given in blade root coordinates. The effective blade length is the projected length onto the root coordinate system in the axial direction. The edgewise/torsion coupling at the initial blade position (blue dashed line) has an opposite direction in comparison to the edgewise/torsion coupling at the deflected blade (red continuous line) position.

ing through moderately large deflections. Hence, the proposed method uses linear mode shapes and linear stiffness, mass and damping matrices for response calculation and quadratic vectors are added to the linear response results to capture large deflection effects. In the studies where intrusive methods are used, `Modal derivatives` are generally preferred as quadratic vectors. For non-intrusive methods, the main purpose is to obtain nonlinear effects from some static solutions and in this study performance of `Expansion modes` are investigated. The formulation and calculation process of these vectors are explained below.

### 3.1 Modal derivatives

The modal derivatives (MDs) are quadratic vectors which include secondary effects that occur due to large deflections (geometric nonlinearities) (Idelsohn and Cardona, 1985). The quadratic relation needs to be written between physical displacements ($\boldsymbol{u}$) and modal amplitudes ($\boldsymbol{q}$) for defining modal derivatives. So, they can be thought of as second derivative of physical displacements ($\boldsymbol{u}$) with respect to modal amplitudes ($\boldsymbol{q}$) when the displacements are represented as a function of modal amplitudes. When Taylor series expansion of the displacements around an equilibrium state ($\boldsymbol{u}_0 = \boldsymbol{u}(\boldsymbol{q}_0)$) are written, the first term ($\boldsymbol{u}(\boldsymbol{q}_0)$) is the equilibrium state, the first derivative represents the linear mode shapes and the second derivative term includes



the modal derivatives as shown in equation (1).

$$
\begin{aligned}
\boldsymbol{u}(\boldsymbol{q}) \approx {} & \boldsymbol{u}(\boldsymbol{q}_0) + \left.\frac{\partial \boldsymbol{u}}{\partial \boldsymbol{q}}\right|_{\boldsymbol{q}_0} \cdot (\boldsymbol{q} - \boldsymbol{q}_0) + \frac{1}{2}\left(\left.\frac{\partial^2 \boldsymbol{u}}{\partial \boldsymbol{q} \partial \boldsymbol{q}}\right|_{\boldsymbol{q}_0} \cdot (\boldsymbol{q} - \boldsymbol{q}_0)\right) \cdot (\boldsymbol{q} - \boldsymbol{q}_0) + \mathcal{O}(\|\boldsymbol{q}\|^3) \\
= {} & \boldsymbol{u}(\boldsymbol{q}_0) + \boldsymbol{\Phi} \cdot \boldsymbol{q} + \frac{1}{2}\left(\frac{\partial \boldsymbol{\Phi}}{\partial \boldsymbol{q}} \cdot \boldsymbol{q}\right) \cdot \boldsymbol{q} + \mathcal{O}(\|\boldsymbol{q}\|^3)
\end{aligned}
\tag{1}
$$

In this study, the equilibrium state is taken as the initial state where $\boldsymbol{u}_0 = 0$ (undeflected state) and equation (1) can be written in terms of mode shapes and their derivatives at the initial state as

$$
\boldsymbol{u}(\boldsymbol{q}) \approx \boldsymbol{\Phi} \cdot \boldsymbol{q} + \frac{1}{2}\left(\frac{\partial \boldsymbol{\Phi}}{\partial \boldsymbol{q}} \cdot \boldsymbol{q}\right) \cdot \boldsymbol{q} + \mathcal{O}(\|\boldsymbol{q}\|^3)
\tag{2}
$$

The linear mode-shapes ($\boldsymbol{\Phi}$) and corresponding natural frequencies ($\omega$) can be found by the generalized eigenvalue solution,

$$
(\boldsymbol{K} - \omega_i^2 \boldsymbol{M})\boldsymbol{\phi}_i = \boldsymbol{0}
\tag{3}
$$

where $\boldsymbol{K}$, $\boldsymbol{M}$ are the tangential stiffness and mass matrices and $\omega_i$, $\boldsymbol{\phi}_i$ are the $i^{th}$ eigenvalue and corresponding eigenvector (mode-shape vector). The stiffness and mass matrices are computed at the initial state ($\boldsymbol{u}_0 = 0$). The modal derivatives are calculated by taking the derivative of eigenvalue problem shown in equation (3) with respect to modal amplitude $q_j$.

$$
\frac{\partial}{\partial q_j}\left((\boldsymbol{K} - \omega_i^2 \boldsymbol{M})\boldsymbol{\phi}_i\right) = \left(\frac{\partial \boldsymbol{K}}{\partial q_j} - \frac{\partial \omega_i^2}{\partial q_j}\boldsymbol{M} - \omega_i^2 \frac{\partial \boldsymbol{M}}{\partial q_j}\right)\boldsymbol{\phi}_i + (\boldsymbol{K} - \omega_i^2 \boldsymbol{M})\frac{\partial \boldsymbol{\phi}_i}{\partial q_j} = 0
\tag{4}
$$

To determine the derivative of $\boldsymbol{\phi}_i$ and $\omega_i$ with respect to the $j^{th}$ modal amplitude $q_j$, another equation is needed. This equation can be chosen as equation (5) where $\boldsymbol{\phi}_i$ is the mass normalized mode shape vector.

$$
\frac{\partial}{\partial q_j}(\boldsymbol{\phi}_i^T \boldsymbol{M} \boldsymbol{\phi}_i) = 2\boldsymbol{\phi}_i^T \boldsymbol{M}^T \frac{\partial \boldsymbol{\phi}_i}{\partial q_j} + \boldsymbol{\phi}_i^T \frac{\partial \boldsymbol{M}}{\partial q_j}\boldsymbol{\phi}_i = 0, \text{ where } \boldsymbol{\phi}_i^T \boldsymbol{M} \boldsymbol{\phi}_i = 1
\tag{5}
$$

When the equation (4) and (5) are combined, the modal derivative of $i^{th}$ mode shape vector $\boldsymbol{\phi}_i$ and natural frequency $\omega_i$ with respect to the $j^{th}$ modal amplitude can be determined by

$$
\begin{bmatrix} (\boldsymbol{K} - \omega_i^2 \boldsymbol{M}) & -\boldsymbol{M}\boldsymbol{\phi}_i \\ -(\boldsymbol{M}\boldsymbol{\phi}_i)^T & 0 \end{bmatrix} \begin{bmatrix} \dfrac{\partial \boldsymbol{\phi}_i}{\partial q_j} \\ \dfrac{\partial \omega_i^2}{\partial q_j} \end{bmatrix} = \begin{bmatrix} -\dfrac{\partial \boldsymbol{K}}{\partial q_j}\boldsymbol{\phi}_i + \omega_i^2 \dfrac{\partial \boldsymbol{M}}{\partial q_j}\boldsymbol{\phi}_i \\ \dfrac{1}{2}\boldsymbol{\phi}_i^T \dfrac{\partial \boldsymbol{M}}{\partial q_j}\boldsymbol{\phi}_i \end{bmatrix}
\tag{6}
$$

Equation (6) contains all the terms that are required to compute the modal derivatives. The derivation of modal derivatives is similar to the derivation of the sensitivity of eigenmodes and eigenfrequencies with respect to a design variable in structural optimization.

For computation of modal derivatives, the terms related to inertia effects (i.e. mass matrix and its derivatives) are generally ignored, since their contribution to the modal derivatives is very limited (Rutzmoser et al., 2017). The derivative of the eigenvalue can also be assumed to be zero due to the fact the eigenfrequencies of the slender cantilever beams are not sensitive to





the vibration amplitude. These assumptions lead to static modal derivatives which are symmetric whereas modal derivatives computed from equation (6) are not necessarily symmetric.

$$\frac{\partial \phi_i}{\partial q_j} = -\boldsymbol{K}^{-1} \frac{\partial \boldsymbol{K}}{\partial q_j} \phi_i \tag{7}$$

For convenience of communication, the static modal derivatives are simply called 'Modal Derivatives (MDs)' hereafter. In equation (7), the derivative of the stiffness matrix with respect to the modal amplitudes, i.e. $\frac{\partial \boldsymbol{K}}{\partial q_j}$, is needed to compute the modal derivatives. In this study, the derivative of stiffness matrix ($\boldsymbol{K}$) with respect to $j^{th}$ modal amplitudes ($q_j$) are computed by central finite difference given in equation (8). The stiffness matrix at the deflected state of the structure by a given amplitudes of $\delta_j$ are computed with a geometrically nonlinear beam solver based on co-rotational formulation in (Krenk, 2005).

$$\frac{\partial \boldsymbol{K}}{\partial q_j} = \frac{\boldsymbol{K}(\phi_j \delta_j) - \boldsymbol{K}(-\phi_j \delta_j)}{2\delta_j} \tag{8}$$

where $\boldsymbol{K}(\phi_j \delta_j)$ is the tangential stiffness matrix when the system displacements equal to $\phi_j \delta_j$. The calculation of $\frac{\partial \boldsymbol{K}}{\partial q_j}$ from equation (8) is the only step where a geometrically nonlinear solver is required. When the stiffness matrix derivatives are ready, the computation of equation (7) is straight forward since the rest of the equation consists of stiffness matrix and linear eigenvectors at undeflected state. The tangential stiffness matrix at undeflected state is actually the linear stiffness matrix which is used in existing modal approaches. There are $M_{MD}$ number of modal derivatives for $M$ number of linear mode-shapes. The relation between $M$ and $M_{MD}$ can be written as,

$$M_{MD} = \frac{M \times (M+1)}{2} \tag{9}$$

Figure 4 and 5 are used to give visual understanding for the modal derivatives. Figure 4 shows bending of a straight beam with airfoil cross-section in one direction and its representation by a linear bending mode and modal derivative of that mode shape with respect to its modal amplitude ($\frac{\partial \phi_i}{\partial q_i}$). The linear bending mode shape doesn't have any displacement in axial direction, so the total beam length increases. The modal derivative vector of the bending mode-shape includes axial displacement effect. When the linear bending mode and its modal derivative is summed, the axial displacement and bending effects are captured together.

Figure 5 shows similar effects with Figure 4 but this time the forces are applied in two bending directions at the same time. Hence, together with axial displacements, there is also torsion effect due to the couplings at deflected state. Modal approach is only capable of capturing the bending deflections in two lateral ($x-$ and $y-$) directions. The axial displacements are captured by the modal derivatives of bending mode shapes with respect to their modal amplitudes as in Figure 4, whereas the torsion effects are included in the cross modal derivative vectors which are derivative of a bending mode-shape with respect to the modal amplitude of a bending mode in other direction ($\frac{\partial \phi_i}{\partial q_j}$).

### 3.2 Expansion modes

Expansion modes are similar to modal derivatives, but they are computed by a non-intrusive approach Hollkamp and Gordon (2008). Expansion modes are computed from the difference (error) between deflections computed by linear ROMs and nonlin-





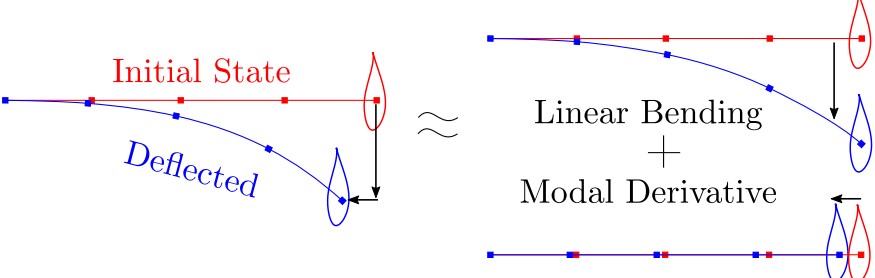

**Figure 4.** Bending deflection of a straight beam and its representation by linear mode shape and its modal derivatives.

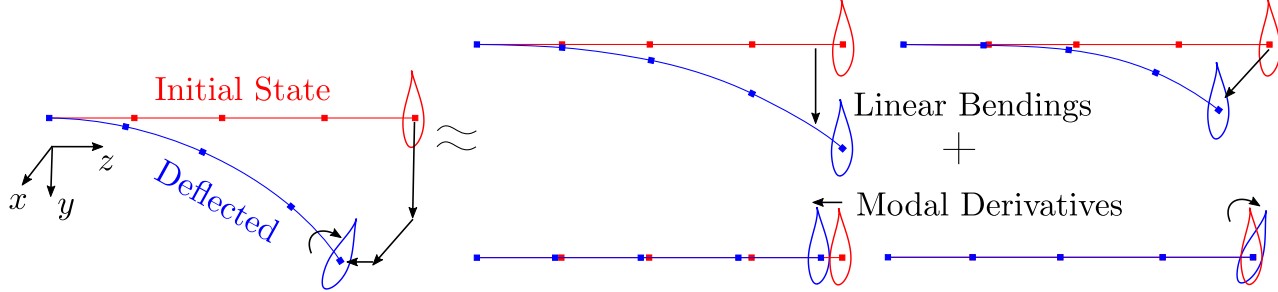

**Figure 5.** Bending deflections of a straight beam in lateral ($x-$ and $y-$) directions and the representation of the deflections by linear mode shapes and their modal derivatives.

ear deflections. The difference is written as a function of selected order of modal amplitudes. In this study quadratic ($2^{nd}$ order)
relation is defined between modal amplitudes and nonlinear deflections. Nonlinear deflections are obtained for static cases by
using a geometrically nonlinear solver. In expansion mode approach, the physical displacements are represented as shown in
equation (10),

$$\boldsymbol{u}(x,t) \approx \boldsymbol{\Phi}(x)\boldsymbol{q}(t) + \boldsymbol{\Phi}_{EM}(x)\,\boldsymbol{q}_E(t) \tag{10}$$

where $\boldsymbol{\Phi}(x)$ and $\boldsymbol{\Phi}_{EM}(x)$ are the matrices whose columns are the linear and expansion mode-shape vectors. The expansion
mode amplitudes $\boldsymbol{q}_{EM}$ which are quadratic functions of bending modes, can be written for $M$ linear mode amplitudes as

$$\boldsymbol{q}_E = \begin{bmatrix} q_1^2 & q_1q_2 & \dots & q_1q_M & q_2^2 & q_2q_3 & \dots & q_{(M-1)}q_M & q_M^2 \end{bmatrix}^T \tag{11}$$

where the $q_i$ is the $i^{th}$ linear mode amplitudes. The relation between number of linear modes ($M$) and number of expansion
mode ($M_{EM}$) corresponding to them can be written as,

$$M_{EM} = \frac{M \times (M+1)}{2} \tag{12}$$

So, there are same number of modal derivatives and expansion mode vectors for the same number of linear modes, since both
of these vectors represents the quadratic relation between deflections and modal amplitudes. The linear mode shape amplitude





can be computed from static ROM formulation,

$$\boldsymbol{\Phi}^T \boldsymbol{K} \boldsymbol{\Phi} \boldsymbol{q} = \boldsymbol{\Phi}^T \boldsymbol{f} = \boldsymbol{K_r} \boldsymbol{q} = \boldsymbol{f_r}$$
$$\boldsymbol{q} = \boldsymbol{K_r}^{-1} \boldsymbol{f_r} \tag{13}$$

where $\boldsymbol{K}$ and $\boldsymbol{K_r}$ are the full and reduced stiffness matrices, and $\boldsymbol{f}$ and $\boldsymbol{f_r}$ are the full and reduced force vectors. When the

applied forces ($\boldsymbol{f}$) are known, $\boldsymbol{q}$ can be computed. The applied forces are chosen as a combination of linear modal shapes ($\boldsymbol{\Phi}$) for quadratic relation of expansion modes and can be determined from,

$$\boldsymbol{f} = \boldsymbol{K}(\lambda_i \boldsymbol{\Phi}_i + \lambda_j \boldsymbol{\Phi}_j) \tag{14}$$

where $\lambda_i$ and $\lambda_j$ are force amplifiers for $i^{th}$ and $j^{th}$ linear mode-shape. Linear modal amplitudes ($\boldsymbol{q}$) and corresponding expansion modal amplitudes ($\boldsymbol{q}_E$) can be computed for $N_k$ number of static solutions from selected combinations of $\lambda_i$ and $\lambda_j$.

The nonlinear displacements ($\boldsymbol{u}$) of these load cases are also computed from co-rotational code. All static load case displacement and modal amplitude results can be collected in matrices $\boldsymbol{U} \in \mathbb{R}^{N \times N_k}$, $\boldsymbol{Q} \in \mathbb{R}^{M \times N_k}$ and $\boldsymbol{Q}_{EM} \in \mathbb{R}^{M_{EM} \times N_k}$ where the columns of the matrices are the each load case displacement and modal amplitude results. The all load case results in matrix form can be written as,

$$\boldsymbol{U}(x) \approx \boldsymbol{\Phi}(x)\boldsymbol{Q} + \boldsymbol{\Phi}_{EM}(x)\boldsymbol{Q}_{EM} \tag{15}$$

Equation (15) includes displacement, modal amplitude and expansion mode amplitude matrices instead of their vector forms in eq. (10). The expansion mode shapes can also be determined by least square method,

$$\boldsymbol{\Phi}_{EM}\boldsymbol{Q}_{EM} = \boldsymbol{U} - \boldsymbol{\Phi}\boldsymbol{Q} \longrightarrow \underset{\boldsymbol{\Phi}_{EM} \in \mathbb{R}^{N \times M_E}}{\text{minimize}} \|\boldsymbol{U} - \boldsymbol{\Phi}\boldsymbol{Q} - \boldsymbol{\Phi}_{EM}\boldsymbol{Q}_{EM}\| \tag{16}$$

where $\| \|$ denotes the 2-norm of the vector and it is solved for Modal expansion matrix $\boldsymbol{\Phi}_{EM}$. Modal expansion vectors are very similar to the modal derivative vectors and they become same (except numeric differences) if the set of $i^{th}$ and $j^{th}$ are

symmetric and force amplitudes $\lambda_i$ and $\lambda_j$ are small enough. The computational cost of expansion modes are more expensive than modal derivatives since their calculation requires more static solutions than modal derivative calculation. Although the difference in computational cost is little for a small number of linear modes ($M$), it becomes clear for a large number of linear modes. On the other hand it is easy to expand the relation between nonlinear displacement and expansion modes so that one can use higher order of relation between modal amplitudes and deflections than second order relations given in equation (11).

### 3.3   Numerical implementation

As mentioned before modal derivatives or expansion mode-shapes are not included into the reduction space and linear mass ($M$), stiffness ($K$) and damping ($C$) matrices are used for the response analysis. This isn't the case for other studies in the literature (see Table 1). Although nonlinear stiffness terms and including the quadratic vectors in reduction space lead to very accurate results, this approach is computationally heavy not only for the computation of nonlinear stiffness terms before time

simulations, also during time simulations where extra iterations are needed due to nonlinear stiffness terms. Since there are




thousands of load simulations required for a wind turbine (Hansen et al., 2015) or aircraft, computational cost increase in time simulations are not desired. Therefore the proposed method is very suitable for load and aeroelasticity analysis due to its simplicity and low computational cost compared to the methods found in the literature.

In this method, modal derivative or expansion mode vectors are used only in post-process step where displacements are computed from modal amplitudes and mode-shape vectors. Algorithm 1 show computation of structural response with quadratic correction vectors. In this study, modal derivative and expansion mode vectors are used as quadratic correction vectors.

---

**Algorithm 1** Response calculation with quadratic correction vectors

---

- Model generation step (time independent and done once)

    1. Generate linear ROM from finite element model:

        (a) Compute mode-shape vectors ($\phi_i$) from $(\boldsymbol{K} - \omega_i^2 \boldsymbol{M})\phi_i = \boldsymbol{0}$

        (b) Compute reduced order stiffness $\boldsymbol{K_r}$, mass $\boldsymbol{M_r}$ and damping $\boldsymbol{C_r}$ matrices by Galerkin projection :
        $$\boldsymbol{K_r} = \boldsymbol{\Phi}^T \boldsymbol{K} \boldsymbol{\Phi}, \, \boldsymbol{M_r} = \boldsymbol{\Phi}^T \boldsymbol{M} \boldsymbol{\Phi}, \, \boldsymbol{C_r} = \boldsymbol{\Phi}^T \boldsymbol{C} \boldsymbol{\Phi}$$

    2. Compute quadratic vectors:

        (a) For modal derivatives see section 3.1

        (b) For expansion modes see section 3.2

- Pure structural response analysis

    1. Compute modal amplitudes for linear ROM by solving equation of motion:
    $$\boldsymbol{M_r} \ddot{\boldsymbol{q}}(t) + \boldsymbol{C_r} \dot{\boldsymbol{q}}(t) + \boldsymbol{K_r} \boldsymbol{q}(t) = \boldsymbol{\Phi}^T \boldsymbol{f}(t) = \boldsymbol{f_r}(t)$$

    2. Compute displacements by using linear mode-shapes, quadratic vectors and modal amplitudes :
    With MDs : $\boldsymbol{u}(x,t) = \boldsymbol{\Phi}(x) \cdot \boldsymbol{q}(t) + \frac{1}{2}\left(\frac{\partial \boldsymbol{\Phi}}{\partial \boldsymbol{q}} \cdot \boldsymbol{q}(t)\right) \cdot \boldsymbol{q}(t)$
    with EMs : $\boldsymbol{u}(x,t) = \boldsymbol{\Phi}(x)\boldsymbol{q}(t) + \boldsymbol{\Phi}_{EM}(x)\,\boldsymbol{q}_E(q)$
    The calculation of displacements ($\boldsymbol{u}(x,t)$) doesn't have to be done at each time step and can be performed after the time simulations.

- Coupled response analysis (time dependent)

    1. Compute modal amplitudes for linear ROM by solving equation of motion:
    $$\boldsymbol{M_r} \ddot{\boldsymbol{q}}(t) + \boldsymbol{C_r} \dot{\boldsymbol{q}}(t) + \boldsymbol{K_r} \boldsymbol{q}(t) = \boldsymbol{\Phi}^T \boldsymbol{f}(t,x) = \boldsymbol{f_r}(t,x)$$

    2. Compute displacements by using linear mode-shapes, quadratic vectors and modal amplitudes :
    With MDs : $\boldsymbol{u}(x,t) = \boldsymbol{\Phi}(x) \cdot \boldsymbol{q}(t) + \frac{1}{2}\left(\frac{\partial \boldsymbol{\Phi}}{\partial \boldsymbol{q}} \cdot \boldsymbol{q}(t)\right) \cdot \boldsymbol{q}(t)$
    with EMs : $\boldsymbol{u}(x,t) = \boldsymbol{\Phi}(x)\boldsymbol{q}(t) + \boldsymbol{\Phi}_{EM}(x)\,\boldsymbol{q}_E(q)$

    3. Update loads ($\boldsymbol{f}(t,x)$) and go to step-1 unless simulation end time is reached

---





## 4    Results and Discussion

The proposed method is tested for a straight beam and IEA15MW wind turbine blade under static and dynamic loads. The displacement results in main loading directions and nonlinear geometric effects especially in axial and torsion directions
are given to evaluate the effects of modal derivatives and expansion modes. The reduced order model results are compared with HAWC2 results for both static and dynamic cases. HAWC2 (Larsen and Hansen, 2019) is a aero-servo-hydro-elastic load analysis tool for wind turbines and developed by DTU Wind and Energy Systems. It uses multibody formulation with Timoshenko beam and can capture geometric nonlinearities (Pavese et al., 2016). In HAWC2, $z-$ axis is along the axial direction whereas $x-$ and $y-$ directions are lateral directions (see Figure 3 for HAWC2 coordinate system). After the results
of test cases, the benefits and limitations of the method is discussed in section 4.3.

### 4.1    Straight beam

A cantilever straight beam model is used for static and dynamic load cases. Table 2 shows the general properties of the beam whose cross-section properties are constant along the beam length and shear coefficients are very high compared to the bending stiffness values, so it behaves like an Euler-Bernoulli beam.

**Table 2.** Straight beam geometry and cross-section stiffness properties. Beam is clamped from its root.

| Length | Unit mass | $EI_{xx}$ | $EI_{yy}$ | $GJ$ |
|---|---|---|---|---|
| $[m]$ | $[kg/m]$ | $[Nm^2]$ | $[Nm^2]$ | $[Nm^2]$ |
| 10.0 | 172.4 | $215\times10^4$ | $869\times10^3$ | $416\times10^4$ |

Figure 6 shows $x-$, $y-$, $z-$ displacement and torsion motion components of the first two mode-shape vectors of the beam. Since the beam is straight and has no material coupling, both bending mode-shapes have only one lateral direction components without any axial or torsion component. First mode-shape is in $x-$ direction and second one is in $y-$ direction.

Figure 7 shows modal derivative (MD) and expansion mode (EM) vectors for first two mode-shapes. The results are very similar for MD and EM vectors. These vectors show the sensitivity of a mode-shape with respect to a modal amplitude. The
vector names are given so that the first number represents the mode number for linear mode-shape vector whose sensitivity is computed with respect to the modal amplitude of the mode number which is given as the second number in the vector names. Hence, MD-1-1, MD-2-2, EM-1-1, EM-2-2 shows the sensitivity of a mode-shape with respect to its own modal amplitude whereas the MD-1-2 and EM-1-2 illustrate the sensitivity of a mode-shape with respect to another mode's modal amplitude. Since, MD-2-1 and MD-1-2 are same for static modal derivatives only MD-1-2 results are shown here. This symmetry case
is also valid for expansion modes (EMs), since they are also computed for static cases. *MD-i-i* and *EM-i-i* vectors have only axial displacements since their mode-shapes are only in one lateral direction. On the other hand, *MD-i-j* and *EM-i-j* have only torsion motions which represents the coupling between two lateral directions.





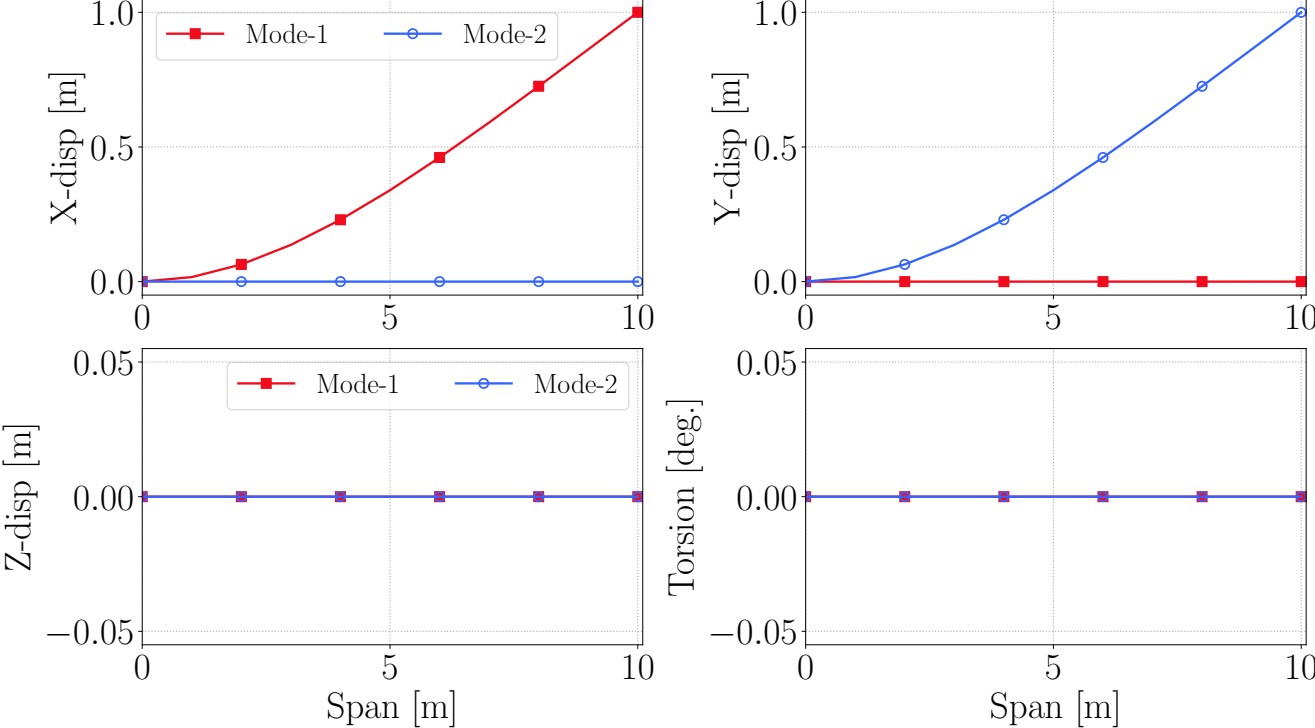

**Figure 6.** $x-$, $y-$, $z-$ and torsion components of first two mode shapes of the straight beam.

The first test was carried with static loads in $x-$ direction. The applied force vectors are determined from stiffness matrix ($\boldsymbol{K}$), first mode shape $\boldsymbol{\Phi}_1$ and amplification factor ($\lambda$) as,

$$\boldsymbol{F}_x = \lambda \boldsymbol{K} \boldsymbol{\Phi}_1 \tag{17}$$

where mode-shape $\boldsymbol{\Phi}_1$ is normalized according to its maximum value.

Expansion mode-shape and modal derivative results are same for this test case, therefore they are given together in Table 3 which shows the tip displacements for linear ROM without corrections (*Lin.*), modal derivative (*MD*) and expansion mode (*EM*) results and HAWC2 results. Modal derivative, expansion mode and linear ROM results are same for lateral ($x-$) deflections due to uncoupled mode-shapes which result in zero values in lateral direction components (see Figure 7). On the other hand, the axial ($z-$) deflection is a secondary effect and captured by correction vectors from modal derivatives and expansion modes. Linear ROM error in $x-$ direction increases as the deflections increases and it is less than $5\%$ for deflections around $20\%$ of beam length. Moreover linear model cannot capture any axial displacement whereas correction factors work quite well.

Figure 8 shows $x-$ and $z-$ (axial) positions of the structural nodes computed from HAWC2, linear ROM, MD and EM models for $\lambda = 2$ and $\lambda = 3$ load cases. The quadratic corrections (MDs and EMs) capture the secondary effects all along the beam length.





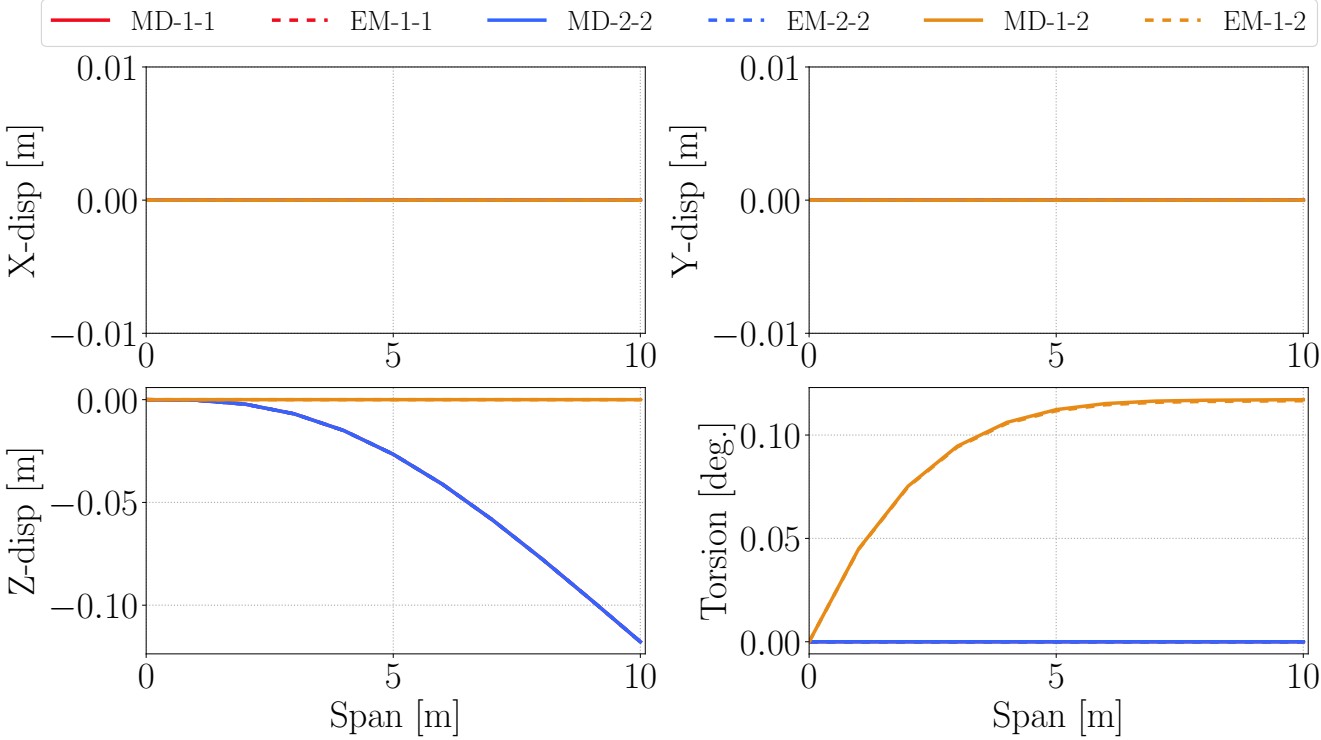

**Figure 7.** $x-$, $y-$, $z-$ and torsion components of modal derivative and expansion mode vectors. MD-1-1, MD-2-2, EM-1-1, EM-2-2 vectors have only axial displacements and MD-1-2 and EM-1-2 vectors have onlyt torsion components.

**Table 3.** Straight beam lateral ($x-$) and axial ($z-$) tip deflection results for HAWC2, linear model, MD and EM corrections.

| $\lambda$ | x-HAWC2 | x-Lin./MD/EM | x-error | z-HAWC2 | z-MD/EM | z-Lin. |
|---|---|---|---|---|---|---|
| 1 | 0.991 | 1.000 | 0.9% | -0.057 | -0.059 | 0.0 |
| 2 | 1.933 | 2.000 | 3.35% | -0.218 | -0.236 | 0.0 |
| 3 | 2.790 | 3.000 | 7.0% | -0.459 | -0.530 | 0.0 |

The second test case includes static loads in $x-$ and $y-$ directions. The load components are determined by similar formulation given in equation (17). The $x-$ force components are amplified by $\lambda = 2.5$ and the $y-$ force components are amplified by $\lambda = 1.0$. Figure 9 shows $x-$, $y-$, $z-$ displacements and torsion deflections along the beam length. Linear model estimates $x-$ and $y-$ displacements quite accurately and quadratic vectors don't alter results in these directions since they don't have components in these directions (see figure 7). On the other hand they capture the secondary effects in $z-$ and torsion directions due to geometric nonlinearities. Linear ROM cannot capture any axial ($z-$) displacement or torsion motion. On the other hand, MD and EM corrections result in accurate estimation of secondary effect even though the loads are applied in two directions. Results show that the quadratic correction vectors can also capture the couplings between different directions.

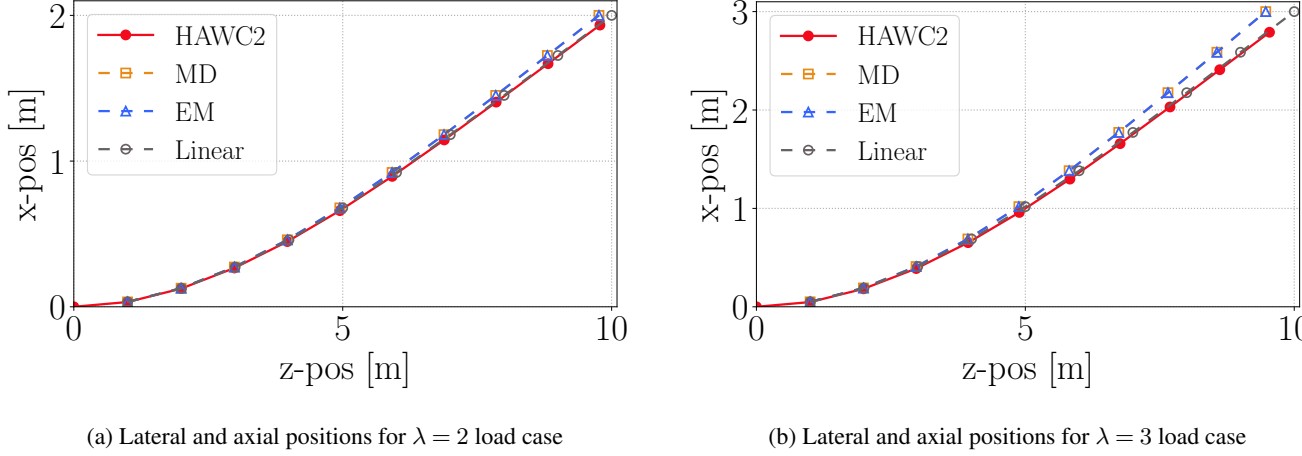

(a) Lateral and axial positions for $\lambda = 2$ load case

(b) Lateral and axial positions for $\lambda = 3$ load case

**Figure 8.** Straight beam $x-$ and $z-$ (axial) positions for $\lambda = 2$ and $\lambda = 3$ load cases. The positions are shown for HAWC2, linear ROM, MD and EM corrections.

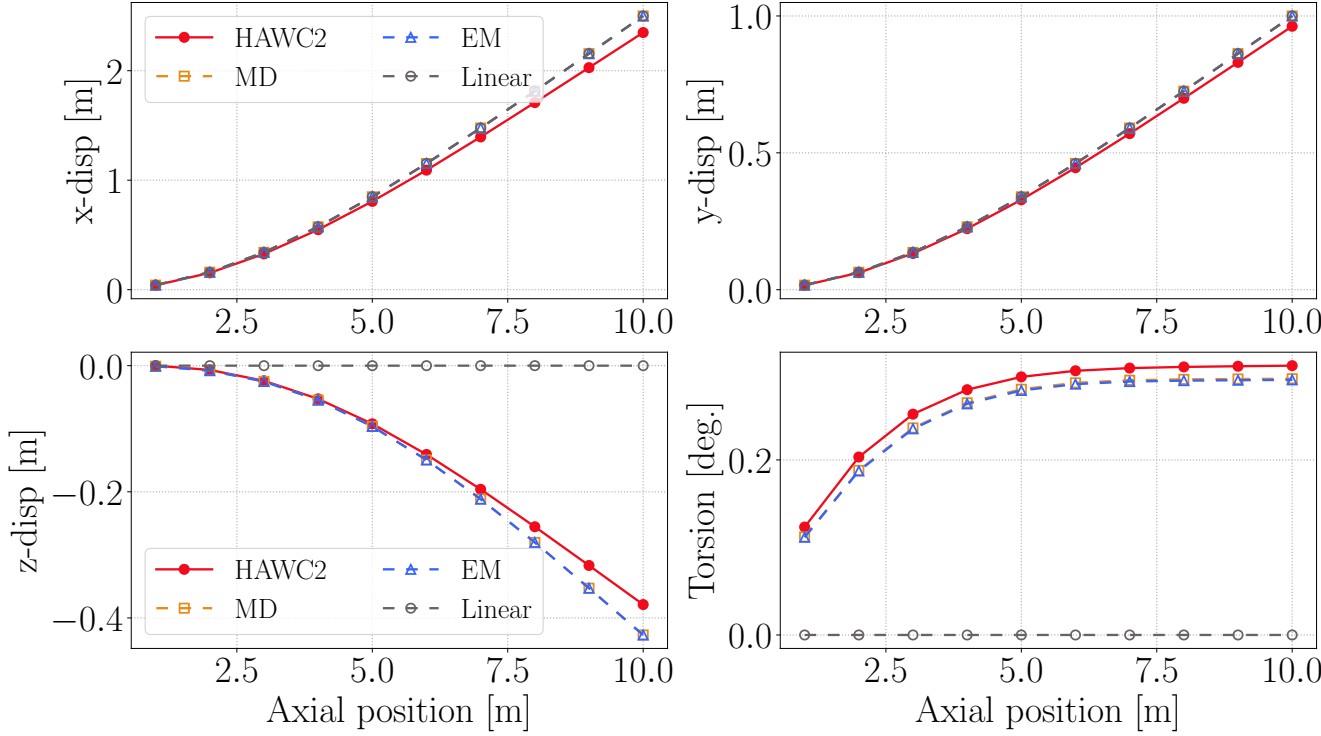

**Figure 9.** Straight beam $x-$, $y-$, $z-$ displacements and torsion motion results along the beam span. The static load has $x-$ and $y-$ components.



WIND
ENERGY
SCIENCE
DISCUSSIONS

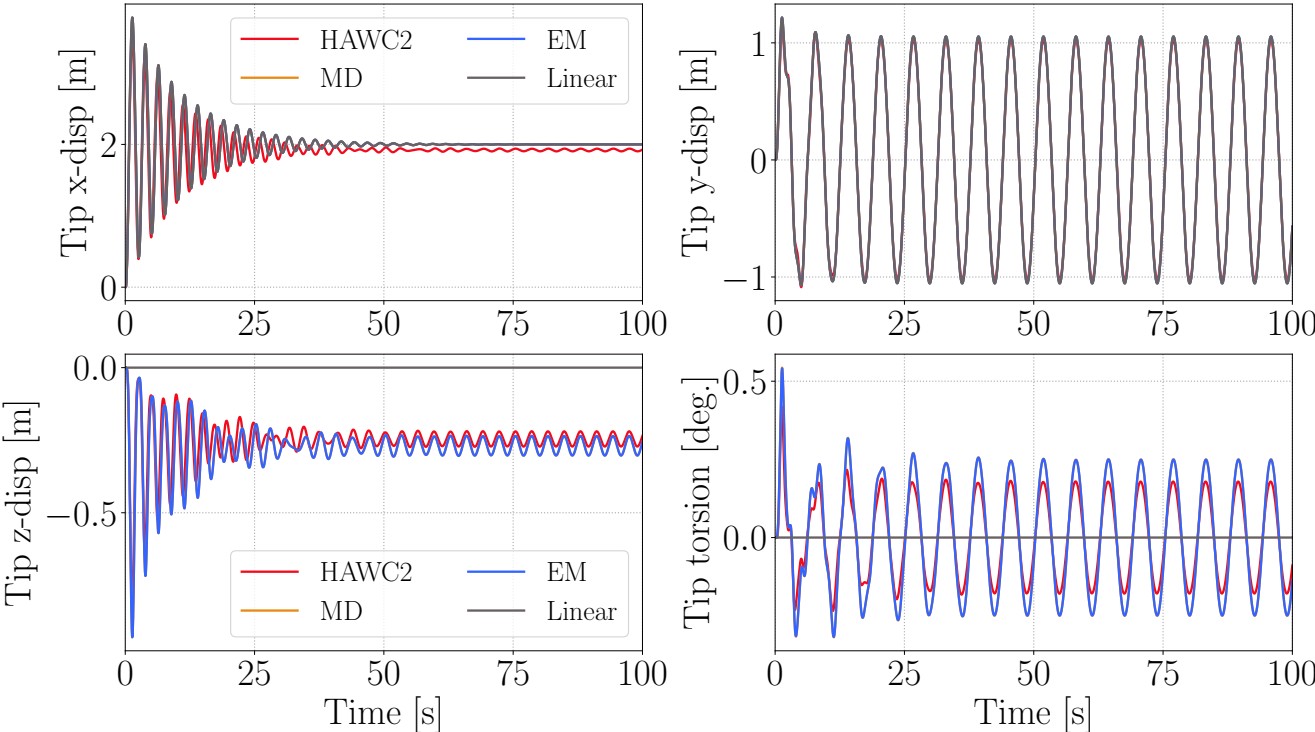

**Figure 10.** Straight beam dynamic analysis tip $x-$, $y-$, $z-$ displacements and torsion deflections for HAWC2, linear ROM, MD and EM corrections.

The last load case with straight beam model includes dynamic loads in two lateral directions. The $x-$ direction load is constant with $\lambda = 2$ whereas the $y-$ direction load is computed as,

$$\boldsymbol{F}_y(t) = \boldsymbol{M}\boldsymbol{g}\sin(\omega t) \tag{18}$$

where $\boldsymbol{M}$ is mass matrix, $\boldsymbol{g}$ is acceleration vector, $\omega$ is the rotation frequency of the beam. For this example the gravity vector has only $y-$ acceleration with value of $9.81\,\mathrm{m\,s^{-2}}$, and $\omega$ is taken as $1\,\mathrm{rad\,s^{-1}}$. The simulation time step is $0.01\,\mathrm{s}$.

Figure 10 shows tip $x-$, $y-$, $z-$ displacements and torsion deflections for $100$ seconds. Linear model can capture dynamics in $y-$ direction very accurately, and its $x-$ direction results are very close to HAWC2 results without any fluctuations after 60 seconds which cannot be captured without nonlinear stiffness terms (Gözcü and Dou, 2020). However these fluctuations are very small compared to the overall displacements in $x-$ direction. On the other hand, linear ROM results are again zero in axial and torsion directions whereas MD and EM corrections work very well even for initial transition period. Moreover, there is no phase difference between HAWC2 and MD/EM results which support that the nonlinear stiffness effects are very small for cantilever beams with moderate deflections. The fluctuations in $x-$ direction due to nonlinear stiffness term effects shows the limitation of this method even though the fluctuations are very small.



## 4.2 IEA15MW wind turbine blade

IEA15MW wind turbine blade is modeled as beams in HAWC2 and FAST for load analysis (Gaertner et al., 2020). It is more
complex and realistic structure in comparison to the beam studied in Section 4.1. The blade has material coupling terms and
initially curved shape due to prebend (mostly in $y-$ direction), aerodynamic twist and backswept (mostly in $x-$ direction).
Hence, the blade mode-shapes are coupled in $x-$, $y-$, $z-$ and torsion directions unlike the straight beam example. Besides, its
cross-section stiffness properties change over the span and material couplings between lateral bending directions and torsion
brings additional couplings in mode-shapes. Table 4 shows the relevant blade properties and details of the design can be found
in (Gaertner et al., 2020).

**Table 4.** IEA15MW turbine blade dimensions, mass and initial curvature limits in prebend $(y-)$ and twist directions.

| Length | Mass | Tip prebend | Max chord | Twist range |
|--------|------|-------------|-----------|-------------|
| $[m]$ | $[ton]$ | $[m]$ | $[m]$ | $[deg]$ |
| 117 | 44.7 | 4.00 | 5.77 | -15 to 2 |

Figure 11 shows first two blade mode-shapes which are coupled due to its geometry and cross-section material couplings.
The first blade mode-shape is mainly in $y-$ (flapwise) direction with $0.5\,\mathrm{Hz}$ whereas the second mode-shape is mainly in $x-$
(edgewise) direction with around $0.7\,\mathrm{Hz}$. Moreover, both mode-shapes have components in all directions including $z-$ (axial)
and torsion directions. Second mode-shape torsion coupling is much stronger than the first mode-shape's coupling whereas it
has weaker coupling in axial direction than the first mode-shape.

Figure 12 shows $x-$, $y-$, $z-$ and torsion components of modal derivative and expansion mode vectors for first two mode-
shapes. MD and EM vectors are very similar and they include components in all directions due to the couplings.

The blade loads include aerodynamic loads at steady $11\,\mathrm{m\,s^{-1}}$ wind speed which gives the highest thrust force for the the
dynamic load case. The steady aerodynamic loads for symmetric rotor (no tilt, now yaw) is time independent (static) and their
torsion components are not included for this example so that the torsion due to mode-shape couplings and dynamic forcing is
more clear for this example. On top of the aerodynamic loads, the weight of the blade is applied in $x-$ (edgewise) direction
with the formulation given in equation (18) where $\omega$ is $1\,\mathrm{rad\,s^{-1}}$.

Figure 13 shows the tip displacement results of the blade in edgewise $(x-)$, flapwise $(y-)$, axial $(z-)$ and torsion directions
for linear ROM, HAWC2 and correction models. Since most of the thrust force is in flapwise direction, the largest displacements
occur in that direction with mean displacement around $13.4\,\mathrm{m}$ for last $50\,\mathrm{s}$. The fluctuations in $y-$ direction is mostly due to
couplings between mode-shapes, in other words it is mostly due to linear stiffness effects and not coming from the nonlinear
stiffness effects which is the case for straight beam example (see Figure 10 $x-$ direction results). Since nonlinear stiffness
effects are very small in $x-$ and $y-$ displacements, the results are very similar in terms of magnitude and phase for all
models. The secondary effects become clear in $z-$ (axial) and torsion directions. The linear model has $0.84\,\mathrm{m}$ mean tip axial
displacement for last $50\,\mathrm{s}$ due to mode-shape couplings (see figure 11), however HAWC2 model has $-0.41\,\mathrm{m}$ mean tip axial

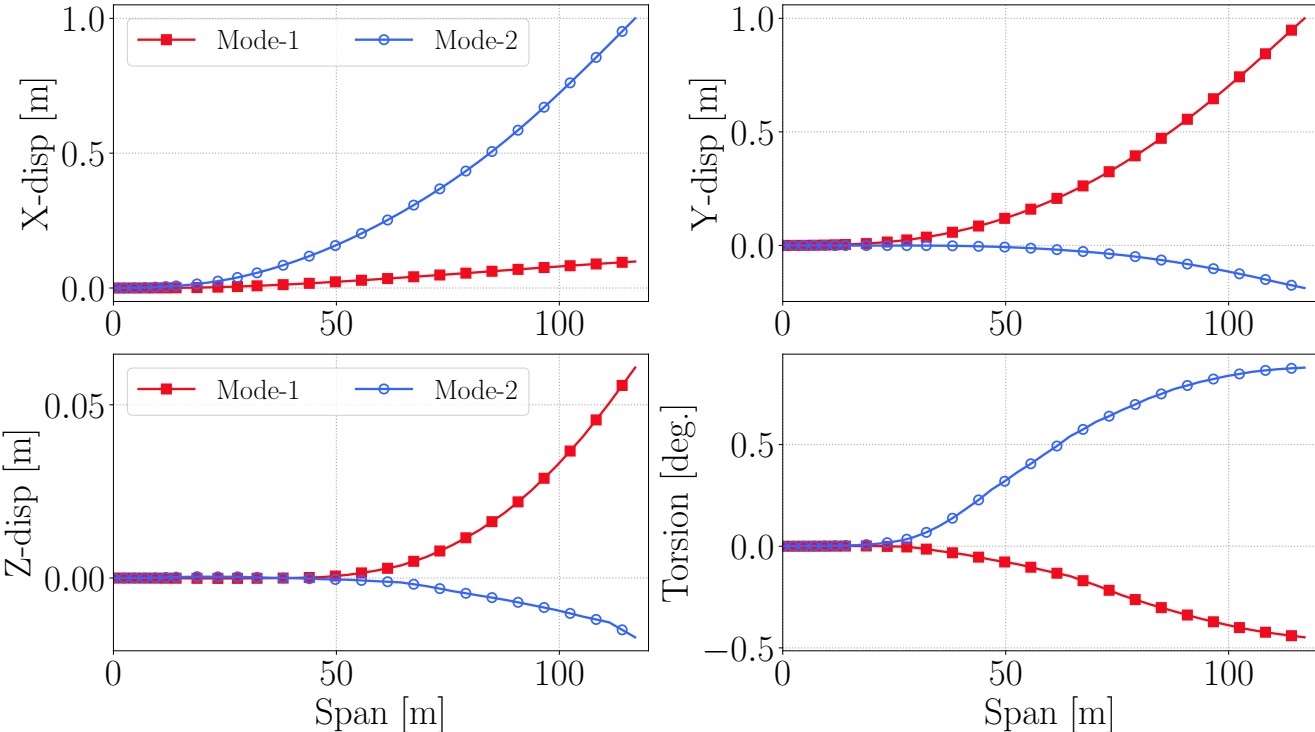

**Figure 11.** $x-$, $y-$, $z-$ and torsion components of first two mode shapes of IEA15MW blade.

displacements for the same time period. Hence, linear ROM estimates $2.5\,\mathrm{m}$ larger rotor diameter compared to HAWC2. Modal derivative correction model has $-0.29\,\mathrm{m}$ and expansion mode correction model has $-0.23\,\mathrm{m}$ mean axial ($z-$) displacements. They also represents the fluctuations in $z-$ directions much more accurately compared to the linear model which has almost no fluctuations for last $50\,\mathrm{s}$. Linear model maximum tip torsion error compared to HAWC2 for last $50\,\mathrm{s}$ is about $1.36°$. The

maximum tip torsion error of MD and EM models for last $50\,\mathrm{s}$ are $0.63°$ and $0.26°$, respectively. The correction modes also capture the torsion motion phase more accurately than the linear model which is out of phase with HAWC2 in torsion direction.

First 15 linear mode-shapes are used in the blade analysis and quadratic vectors for first three modes are included since these modes have very high modal amplitudes among all. Average modal amplitude values of first seven mode-shapes for the last $50\,\mathrm{s}$ of dynamic analysis are given in Table 5. Other mode-shapes have much less average values than $0.2$.

**Table 5.** IEA15MW blade mean modal amplitude results for last $50\,\mathrm{s}$ of the analysis

| $q_1$ | $q_2$ | $q_3$ | $q_4$ | $q_5$ | $q_6$ | $q_7$ |
|-------|-------|-------|-------|-------|-------|-------|
| 12.44 | 1.13 | 0.74 | -0.17 | 0.23 | 0.28 | 0.34 |





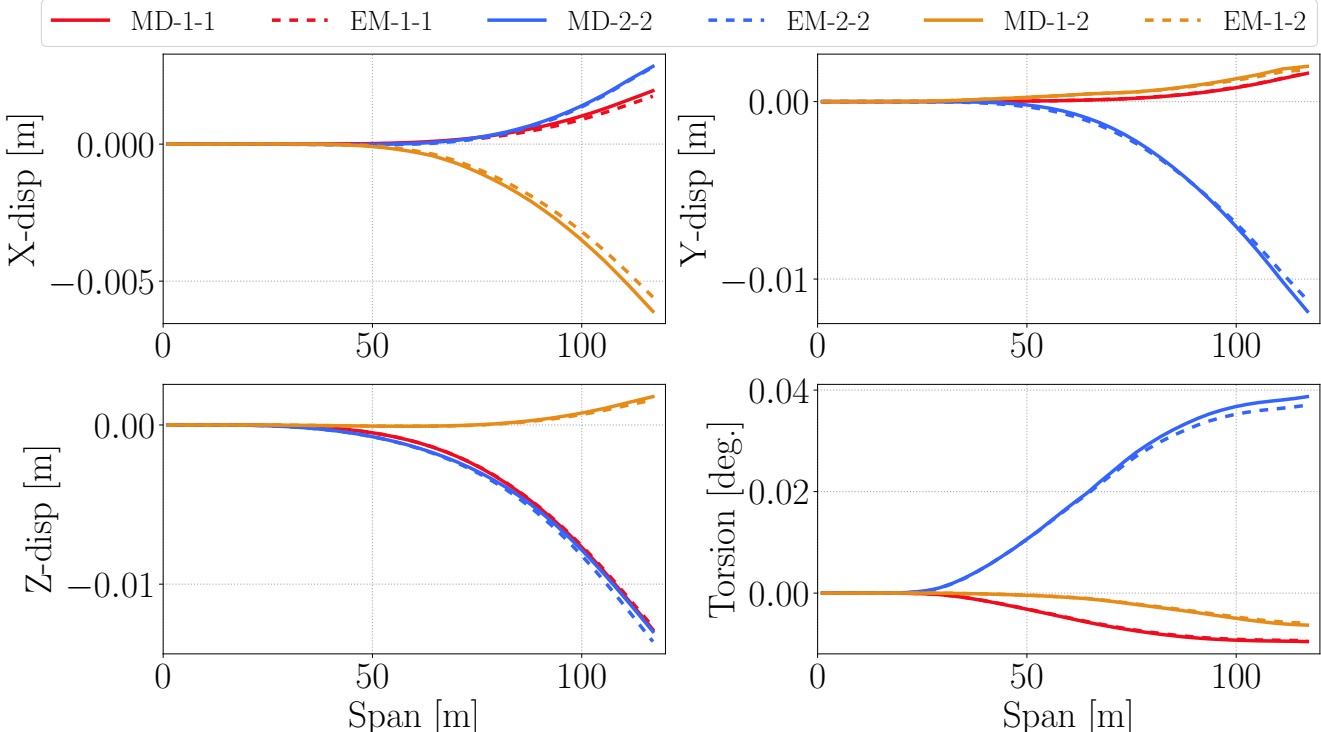

**Figure 12.** $x-$, $y-$, $z-$ and torsion components of modal derivative and expansion mode vectors

Figure 14 and figure 15 show the displacement results over the blade span at the two selected time steps where minimum and maximum torsion deflections are obtained in the last $50\,\mathrm{s}$ from HAWC2 (see Figure 13). These time steps also correspond to maximum and minimum edgewise displacements. Figure 14 shows displacement results along the blade span at $67.51\,\mathrm{s}$ where the maximum tip torsion occurs. Linear ROM results are quite accurate except $z-$ and $x-$ displacement. On the other hand, MD and EM results are much better than linear ROM results in all directions. Linear ROM has $1.2\,\mathrm{m}$ error in $z-$ direction at blade tip whereas MD and EM errors are less than $0.17\,\mathrm{m}$ for blade tip.

Figure 14 shows displacement and torsion results along blade span at the maximum tip torsion moment ($70.56\,\mathrm{s}$ in Figure 13). Linear ROM is quite accurate in $x-$ and $y-$ direction, however it has low accuracy in $z-$ and torsion directions. Its total torsion error over the blade span is $33.6°$ whereas MD error is $7.7°$ and EM error is $1.7°$ along the blade span compared to HAWC2 results. The $z-$ direction results are similar to minimum torsion moment results in Figure 15.

One of the findings in Figure 10 and Figure 13 is that the frequency of the vibration is not affected by the large deflections of the blade. Even the linear model can predict the vibration frequency or the time period of the periodic response in good agreement with HAWC2 simulation in main deflection directions. In practice, this means that no correction is needed for the vibration frequency in case of large deflections. This also suits the application of the proposed correction method to a variety of slender cantilever structures including wind turbine blades. Besides, this allows using correction terms to capture secondary





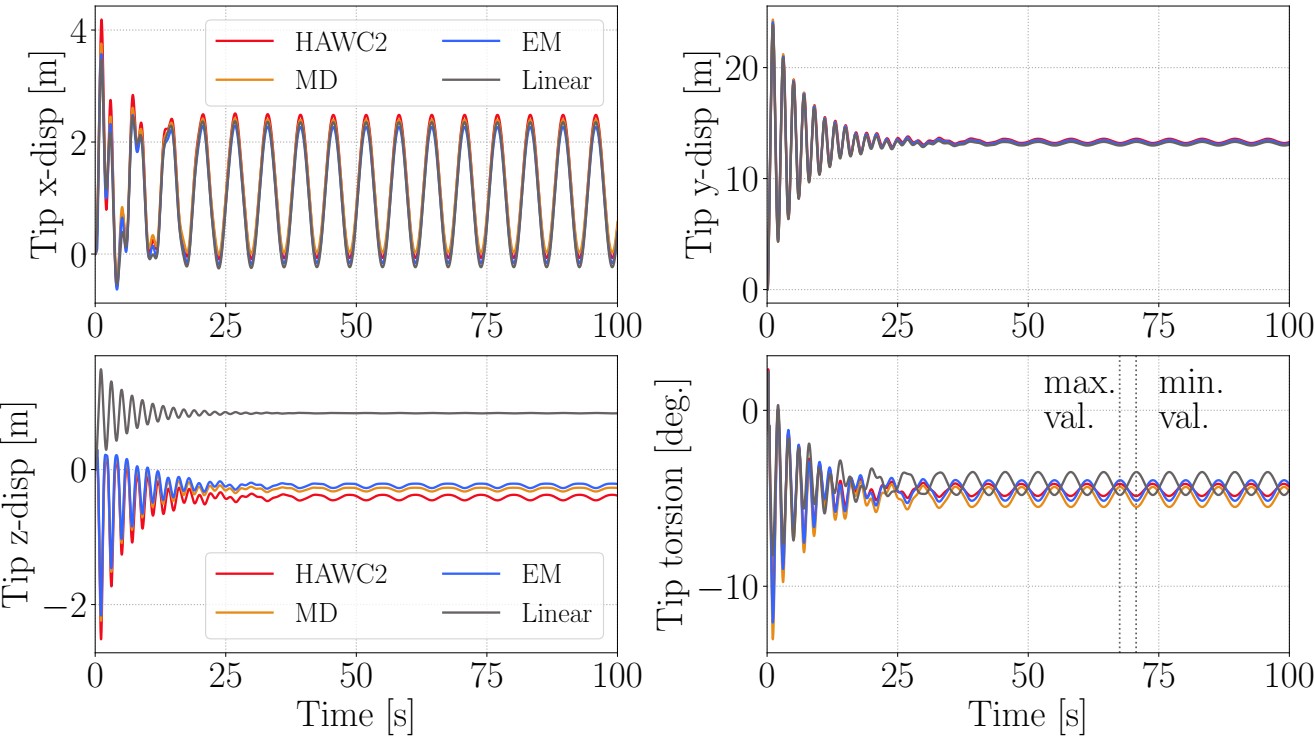

**Figure 13.** IEA15MW blade tip $x-$, $y-$, $z-$ displacements and torsion motion results for HAWC2, linear ROM, MD and EM corrections.

effects due to geometric nonlinearities for moderately large deflections. On the other hand, linear ROM has wrong phase in secondary deflections directions such as axial and torsion directions whereas correction terms correct the phase and magnitude of deflections in those directions.

In this study, the load time series are prescribed and not updated with deflection. In other words, the results are not obtained for an aeroelastic analysis. It is known that the importance of torsion motion will increase for an aeroelastic analysis. The

difference in mean torsion results are critical in static aeroelastic analysis where the steady operating conditions are determined. On the other hand, the difference in torsion fluctuations will affect the damage equivalent loads Gozcu and Verelst (2019) and aeroelastic stability analysis Kallesøe (2011).

### 4.3 Benefits and Limitations

Results show that the proposed method improves the accuracy of linear cantilever beam models with simple corrections es-

pecially in axial and torsion directions in which the secondary effects are more apparent. Moreover, results corroborate the assumption that the nonlinear stiffness effects in bending directions are not very critical for cantilever beams undergoing moderately large deflections. This is shown by figures 10 and 13 and seems to be valid for the cases studied in this work (deflections reaching up to 25 % of the beam length). So, the method captures secondary effects in axial and torsional directions due to



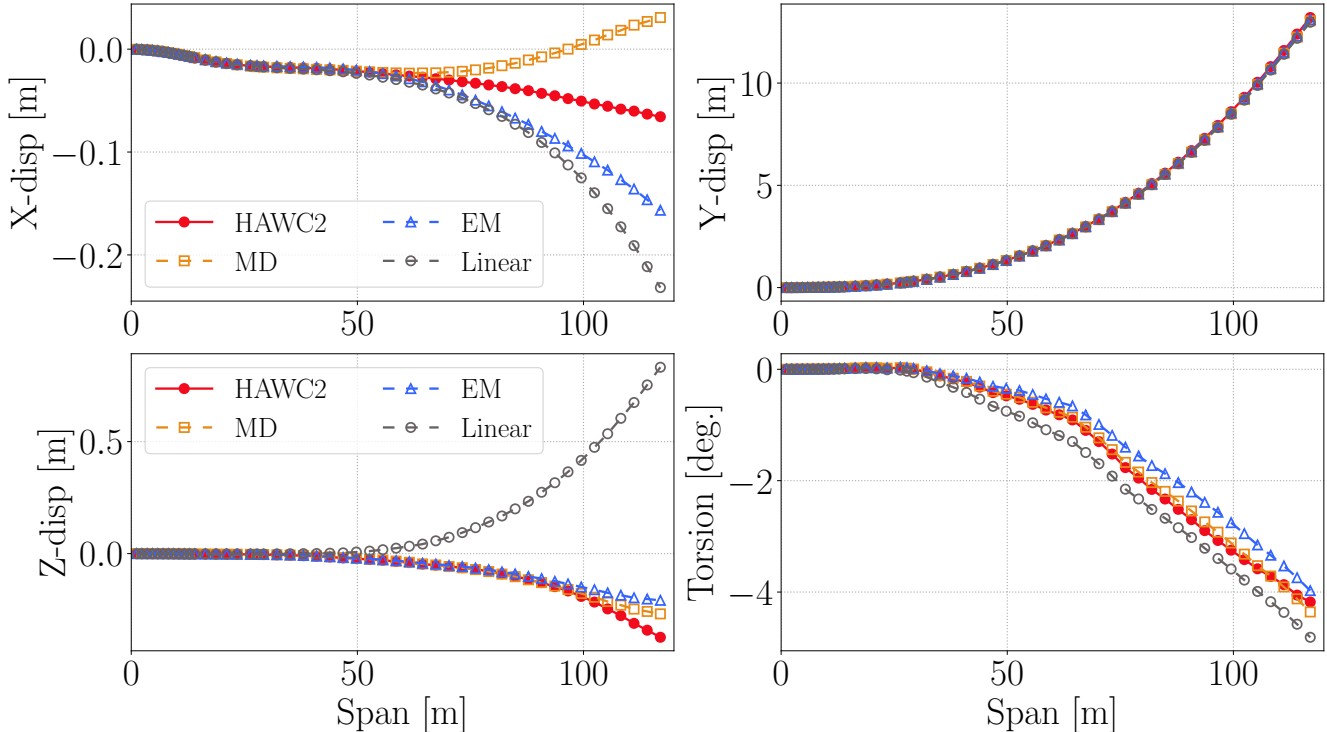

**Figure 14.** Spanwise deflection results at the time when the maximum torsion occurs 67.51 s

bending deflections when the linear bending deflections are captured accurately enough. Because the method is based on linear
vibration modes and modal derivatives (or expansion modes), it is suitable for deformation and vibration up to a moderate
level.

Although the method allows capturing geometrically non-linear effects for the given examples, it doesn't include any non-
linear stiffness, damping or inertia terms in response computation. Hence it shouldn't be considered as a general geometrically
nonlinear method. For instance, small fluctuations in $x-$ bending direction for straight beam dynamic case (see figure 10)
cannot be captured by the method since these dynamics require nonlinear stiffness terms in response computation. Moreover,
it is intended for cantilever beam-like structures, and thus not suitable for structures such as clamped - clamped beam-like
structures.

## 5 Conclusions

This study introduced a correction method for capturing geometric nonlinear effects for reduced order models of cantilever
beams which go through moderately large deflections. The method includes quadratic correction vectors which includes nonlin-
ear effects. Two different quadratic vectors, modal derivatives and expansion modes, were investigated in this study. Correction

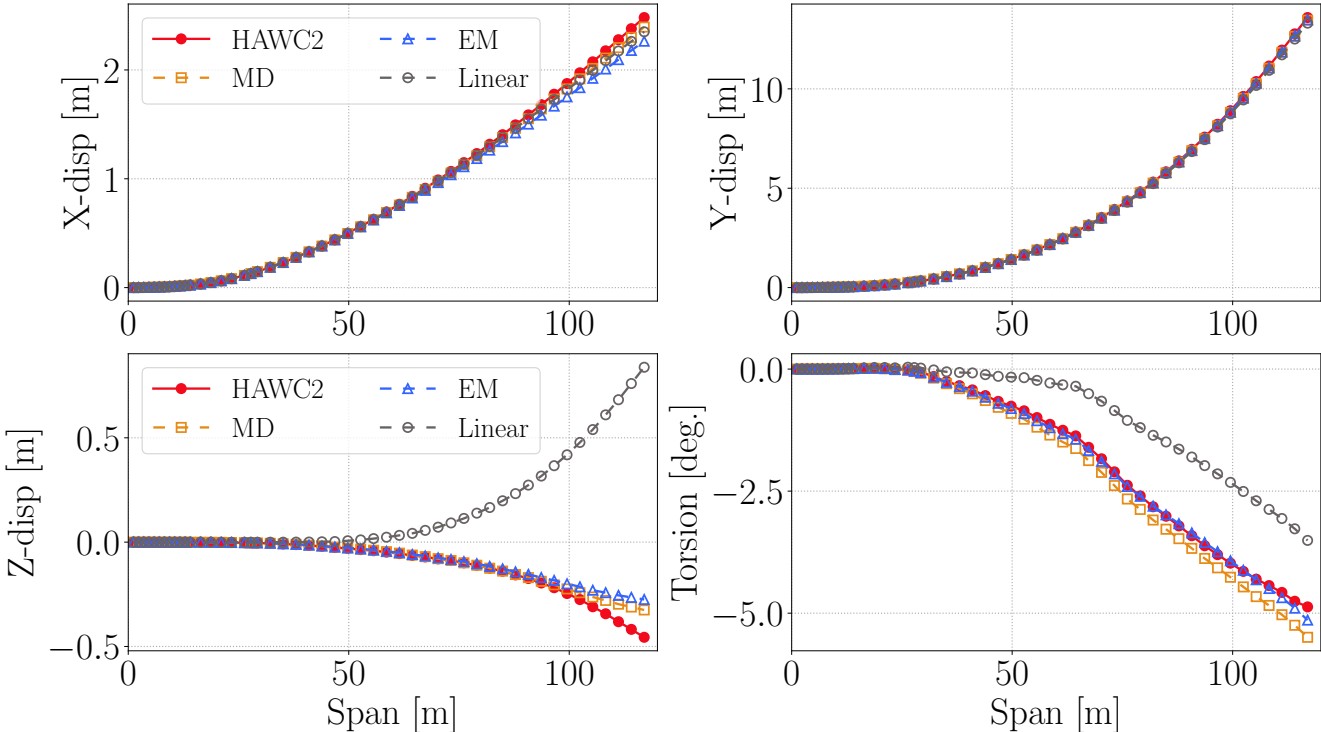

**Figure 15.** Spanwise deflection results at the time when the minimum torsion occurs 70.56 s

vectors for both methods are computed once by using a geometrically nonlinear beam solver. Therefore their computation cost becomes insignificant when it is compared to the computational cost of thousands aircraft wing or wind turbine blade aeroelastic simulations. The advantage of this method is its speed and simplicity, and it is considered as a quite convenient correction

for aeroelastic analysis tools which uses modal based reduced order models for aircraft wings or wind turbine blades.

The proposed method is tested with a straight beam which has uncoupled mode-shapes and IEA15MW turbine blade which has coupled mode-shapes due to its geometry and cross-section stiffness matrix. In the straight beam example, modal derivatives and expansion modes have very similar results and they captured geometrically nonlinear (secondary) effects very accurately for lateral deflections up to 25 % of beam length. IEA15MW blade results showed that linear ROM axial results are in

opposite direction compared to HAWC2 results and it estimates 1.25 m longer blade than HAWC2. Correction models have very good estimation for axial displacements of the blade. Besides, linear model torsion results are out of phase with HAWC2 results where correction models are in-phase. These results show that the proposed corrections increases accuracy of cantilever beam reduced order models significantly especially for structures with coupled mode-shapes. On the other hand, the method is not suitable for the cases where the nonlinear stiffness or inertia or damping terms have substantial effect on response since it

uses only linear terms in response computation.





The study can be expanded with implementation of the correction terms into an aeroelastic tool which uses reduced order models for airplane wings or wind turbine blades. Alternatively, the proposed correction methods can also be implemented based on machine learning approaches such as neural network models or adaptive kiriging methods.

*Author contributions.* O.G. developed the theoretical formalism and carried out the numerical implementation. E.B and S.D. helped shaping
the content of the paper and tests cases.

*Competing interests.* DTU Wind Energy develops, supports and distributes HAWC2 on commercial terms.

*Acknowledgements.* The study was co-financed by Ørsted A/S and DTU Wind and Energy Systems Department.



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
