# Peer review of "A correction method for large deflections of cantilever beams with modal approach"

_Wind Energy Science, 2022_

## Referee Comment (RC2)

The authors present two methods to compute second-order expansion of the displacement field in reduced-order models. The work is relevant, because, as mentioned by the authors, reduced order models are still needed, despite the existence of geometrically exact beam theories. Further, references on the topic are often hard to access or understand, and the topic on how to determine the second-order terms is not always well explained.

I have several general comments that I hope can improve the paper:

- The authors start with a nice table summarizing some work done in the literature. I think the authors should slightly expand on what the possible options are when they present the table, distinguishing between what is meant by intrusive and nonintrusive, what are the von Karman kinematics, what are dual modes, which method use modal derivatives or expansion modes. Later in the text (Section 3.3), the authors mention "nonlinear stiffness terms", and "including quadratic vectors in the reduction space". These notions likely need to be introduced early on (and maybe included in the table), to give a better context on the method presented by the authors, and what the alternatives are.

- When reading the paper, it was not clear to me what the contribution from the author was, so I would recommend making it clearer in the text (as mentioned in the point above, presenting a broader context could help). I can be wrong,  but I believe that second-order expansions of the displacement field are already presented in the literature  (and with a strong coupling to the equation of motions, i.e. including centrifugal stiffening and the like), and both the modal derivatives and the expansion modes method are already developed in the literature (please correct me if I'm wrong). Even if that's the case, I think it's nice that the paper presents both methods in a nice uniform way and compares them against HAWC2.

- If my understanding is correct, the second-order expansion has no stiffness, damping or inertial influence on the system. It's only used as a postprocessing step, to compute the displacement field u, based on q. Based on u, different external loads may be obtained, but the stiffness, damping or inertial influences of the "extra displacement" are not captured. This would mean that effects such as centrifugal stiffening or tower softening, would not be captured by the method. If I'm right (or even if I have misunderstood it) I think the paper should discuss this more clearly in the introduction and potentially in a discussion section.

- The paper contains several typo/editorial issues. I have corrected some of them up until the beginning of section 4. I would advise a thorough reading of the last sections.

I enclose some specific comments below. I hope that addressing my general and specific comments will improve the paper. Despite my comments, I'd like to congratulate the authors for their interesting and thorough work. I'll be looking forward to review a revised version of this paper.

   Emmanuel

p1 l12: is used > are used

p1 l24: (for your information, the ElastoDyn module of OpenFAST contains corrections for geometric nonlinearities. Unfortunately, this is not published.)

p2 l 29: You may consider adding the following (either in the table or simply in the literature review above):

- https://link.springer.com/chapter/10.1007/978-94-017-0625-4_33

- https://doi.org/10.1002/we.2327  (Section 4.2, 4.3)

- https://doi.org/10.5194/wes-2021-46 (Appendix C and D3-D5, I will communicate these separately)

It would be interesting to put the method presented in these references in perspective with your approach.

p2 l34: The proposed ->  Our proposed

p3 l36: If possible, you could add more background on what is meant by intrusive and nonintrusive.  You could also add small explanations for the different methods and reduction bases presented in table 1. As mentioned in a later comment, the term "expansion modes" may be introduced here as well. Alternatively, point to later sections where they are described.

p3 l46: Most of the ... : I'm not sure this is fully true, I would skip or reformulate this statement.

p3 l49: is called as -> is referred to as ""

p3 l49: in case -> in the case

p3 l50: a* lateral deflection

p3 l51: in *the beam span direction

p3 l54-55: "for instance [...]" : this sentence could be reformulated to be more precise.

p4 l58:  but also in the* torsional* direction

p4 l58:  load is* applied

p4 l65:  split the sentence: "in the figure. However, ..."

p4 l66: change sign through flapwise bending

p4 l66: alters blade loads and aseroelastic stability

p4 l73: It might be relevant to also cite the work of Wallrapp here, and put your work in perspective. Wallrapp also uses a Taylor expansion for the displacement field. For instance, the following references could be relevant: https://doi.org/10.1002/nme.1620320818, https://link.springer.com/chapter/10.1007/978-94-017-0625-4_33, http://dx.doi.org/10.1080/08905459408905214

p4 l73: The term "Expansion modes" is not present in Table 1. Maybe it would make sense to introduce it in the table as well.

p5 l76: remove "through"

p6 eq 3: This eigenvalue problem assumes that the mass and stiffness matrices are constant (taken as the undeflected position). Maybe you could add some precision in the text. Would the method need to be modified if centrifugal stiffening effects are to be included? Would different shape functions be generated for different rotational speed?

p6 l 97: Can you clarify what is meant by "tangential" in this sentence?

p7 l116: for convenience of communication -> "for conciseness"

p7 l117: the derivative of the* stiffness matrix wrt the* jth

p7 l118: finite differences*, as* given  in eq 8

p7 l120: is* computed

p7 l124: straightforward*

p7 l125: at the* undeflected

p7 l130: understanding of* the modal

p7 l135: the paragraph needs many grammatical corrections.

p7 l142: parenthesis are needed before the citation

p8 l144. In this study,* a*

p8 l146. In the* expansion mode

p8 eq10: the variable x is not introduced in the text and might not correspond to the one used in Figure 4&5.

p8 l 153: replace with: and the number of corresponding expansion modes (M_EM) is:

p9 l155-170. The procedure here is hard to follow. I would recommend rewriting it to make it clearer.

    - It is not clear to me why the amplitudes need to be computed based on a static formulation (with no inertial contributions).

    - I'm guessing you are describing a procedure to obtain Phi and Phi_EM based on some "unit" static force fields.

    - You might want to address ealier in the text how Phi is determined in the expansion modes method (probably the same way as the modal derivative method, using eq 3).

    - How is Phi_EM determined in equation 15? I'm understanding that for a given force field, u is obtained from a nonlinear code,

    and then q and q_em are obtained by identification. Is that the case? But that means that Phi and Phi_EM needs to be known somehow, no?

p9 l171: You mention that Phi_EM can "also" be determined using least square, but it is not clear what this "also" means here.

It was not clear in the lines above how Phi_EM was determined.

If there is indeed two ways to determine Phi_EM, you could use two separate subsections to distinguish the two methods.

But most likely, equation 16 is actually the main method you are using to determine Phi_EM.

As you can see, the reader can get quite confused, so I would advise to revise this section, trying to guide the reader more. For instance, stating earlier "The procedure to obtain Phi_EM is as follows. First, we do X. Then Y".

p9 l174: Correct the sentence to: The modal expansion vectors are similar to the modal derivative vectors. They are identical (except for numerical differences) when ...

p9 l175: are more -> is more

p9 l177: little-> small, clear -> significant

p9 l178: On the other hand,* (comma)

p9 l179: this sentence should preferably be reformulated

p9 l181: as mentioned before, (comma)

p9 l181: this sentence is not clear. In general, you might want to expand this paragraph a little, to explain what is different in the quadratic correction vector method.

You probably need to mention somewhere in the text (maybe early on), what is meant by "nonlinear stiffness terms", and "including quadratic vectors in the* reduction space".

For instance, it was not clear to me why you mentioned "post-processing" on line 190. (the Phi* are computed as preprocessing steps, but the displacement field can indeed be computed as a postprocessing step).

p10 l 190: shows* the* procedure to obtain the structural response with

p10 Algorithm: It seems that the Taylor expanded field does not introduce additional forces to the system because it only appears as a postprocessing step, which influences the displacement field (and thereby the external loads), but not the stiffness or inertia of the system. This is an important assumption that should be mentioned early on and discussed further.

p10: "Coupled response analysis (time-dependent)". "Time-dependent" is not clear here. In the "Pure structural response", the force was a function of time as well. I believe what is meant here is that the loads are nonlinear functions of q (or x)

p11 l193: and the* IEA 15-MW

p11 l194-195: the sentence is not clear

p15 eq 18: It is mentioned that "omega" is the rotation frequency of the beam. Is the beam rotating, or is it only the loads that are fluctuating with a given frequency?  How are centrifugal stiffening effects accounted for if the beam is rotating?

end of p15, and p21 l338: You observe that MD and EM have very similar results. My understanding is that these results be identical in theory, no? The only difference here comes from the numerical method used to determine the first-order terms in the preprocessing step. I would recommend adding a small paragraph at the end of this result section to mention that (if I'm correct) or discuss it (if I'm wrong).

p18 l295: It seems the linear ROM predicts an elongation, whereas the other methods predict a compression (or the opposite). Could you discuss the reason for these differences here?

  (Also, and that is my misunderstanding, why is the linear ROM predicting a displacement in the z-direction? Is there a shape function in the elongation direction? It might be worth mentioning for other readers.)

p21 l340: I think the results about 1.25m longer blade was not mentioned earlier in the results section (you might have used 1.2m), and I think this sentence is a bit unclear.

---

## Author Comment (AC1)

**Response to the reviewers on the manuscript "A correction method for large deflections of cantilever beams with modal approach", by Ozan Gözcü, Emre Barlas, and Suguang Dou.**

The authors thank the two reviewers for their valuable and useful feedback. The comments and suggestions have certainly helped to improve the quality of the manuscript. We have attempted to address all points raised in the reviews and hope that the replies and changes to the revised manuscript are satisfactory.

Below, the detailed comments from the two reviewers are addressed with the following color code.

The reviewer's comments are written in black text.

Author's responses are written in blue text.

Changes in the paper are outlined in green. The relevant text is also copied here for convenience of re-review.

Best regards,

The authors

**Reviewer #1:**

A good and clearly presented paper. An interesting prospect to enhance modal based codes.

The only dynamic cases shown use MD and EM corrections based on the mean deflection in the dynamic simulation (e.g. the IEA 15MW deflected by aerodynamic forces corresponding to 11m/s wind). It would be important to check how well the method works for a case where the flapwise deflection can change from positive to negative e.g. in a turbine shutdown event. Correction terms based on only one modal amplitude might not work so well then. However, these limitations are alluded to by the author around line 310.

The authors thank the reviewer for suggesting this interesting case. The proposed method in this study is capable to handle this case with satisfactory accuracy. The negative flapwise deflections have a similar effect to prebend in terms of torsion - edgewise coupling and as shown in IEA15 MW blade example for the proposed method can correct linear model results and reduces the error compared to the nonlinear results. The method is implemented into Orsted's FLEX tool and results showed that it works as expected. We are planning to publish another paper with FLEX implementation.

Reply to reviewer's suggested minor changes / clarifications:

Figure 7 – not all of the lines on the plot are visible

Thank you for the review. Markers are added to the lines in Figure - 7 and Figure - 12.

Figure - 7 and Figure - 12

Table 3 – why no % error presented for Z deflection?

The authors thank the reviewer for the suggestion. The percentage errors were added and the comments were updated about the results for z displacement.

Table - 3 is updated and the text before the table is updated according to the new columns.

**Reviewer #2:**

The authors present two methods to compute second-order expansion of the displacement field in reduced-order models. The work is relevant, because, as mentioned by the authors, reduced order models are still needed, despite the existence of geometrically exact beam theories. Further, references on the topic are often hard to access or understand, and the topic on how to determine the second-order terms is not always well explained.

The authors start with a nice table summarizing some work done in the literature. I think the authors should slightly expand on what the possible options are when they present the table, distinguishing between what is meant by intrusive and nonintrusive, what are the von Karman kinematics, what are dual modes, which method use modal derivatives or expansion modes. Later in the text (Section 3.3), the authors mention "nonlinear stiffness terms", and "including quadratic vectors in the reduction space". These notions likely need to be introduced early on (and maybe included in the table), to give a better context on the method presented by the authors, and what the alternatives are.

The authors thank the reviewer for the useful comments and suggestions. We have added text to introduce intrusive, non-intrusive, and von Karman Kinematics. von Karman Kinematics is suitable for describing the membrane effect or bending-extension coupling.

Introduction content is updated.

When reading the paper, it was not clear to me what the contribution from the author was, so I would recommend making it clearer in the text (as mentioned in the point above, presenting a broader context could help). I can be wrong, but I believe that second-order expansions of the displacement field are already presented in the literature (and with a strong coupling to the equation of motions, i.e. including centrifugal stiffening and the like), and both the modal derivatives and the expansion modes method are already developed in the literature (please correct me if I'm wrong). Even if that's the case, I think it's nice that the paper presents both methods in a nice uniform way and compares them against HAWC2.

Thanks for the review, we updated introduction regarding to your review.

Main contributions of the study is listed at the end of the introduction.

If my understanding is correct, the second-order expansion has no stiffness, damping or inertial influence on the system. It's only used as a postprocessing step, to compute the displacement field u, based on q. Based on u, different external loads may be obtained, but the stiffness, damping or inertial influences of the "extra displacement" are not captured. This would mean that effects such as centrifugal stiffening or tower softening, would not be captured by the method. If I'm right (or even if I have misunderstood it) I think the paper should discuss this more clearly in the introduction and potentially in a discussion section.

Thanks for the review, we updated 'relevant kinematics' section regarding to your review.

See section-2 (relevant kinematics) section for changes.

The paper contains several typo/editorial issues. I have corrected some of them up until the beginning of section 4. I would advise a thorough reading of the last sections.

p1 l12: is used → are used

Thanks, it is corrected.

p1 l24: (for your information, the ElastoDyn module of OpenFAST contains corrections for geometric nonlinearities. Unfortunately, this is not published.)

There was a presentation which shows tip axial displacement corrections but I couldn't find it again.

p2 l 29: You may consider adding the following (either in the table or simply in the literature review above): `https://link.springer.com/chapter/10.1007/978-94-017-0625-4_33` `https://doi.org/10.1002/we.2327` (Section 4.2, 4.3) `https://doi.org/10.5194/wes-2021-46` (Appendix C and D3-D5, I will communicate these separately) It would be interesting to put the method presented in these references in perspective with your approach.

Thanks for the resources. Branlands' 2019 Flex paper and review paper from 2022 are cited in Section-2 for their multibody and elastic body models and centrifugal stiffening corrections.

p2 l34: The proposed → Our proposed

Thanks, it is corrected.

p3 l36: If possible, you could add more background on what is meant by intrusive and nonintrusive. You could also add small explanations for the different methods and reduction bases presented in table 1. As mentioned in a later comment, the term "expansion modes" may be introduced here as well. Alternatively, point to later sections where they are described.

Thanks for the comment. Introduction section is updated regarding to your review.

See 'Introduction' section

p3 l46: Most of the ... : I'm not sure this is fully true, I would skip or reformulate this statement.

Thanks, it is corrected.

p3 l49: is called as → is referred to as ""

Thanks, it is corrected.

p3 l49: in case → in the case

Thanks, it is corrected.

p3 l50: a* lateral deflection

Thanks, it is corrected.

p3 l51: in *the beam span direction

Thanks, it is corrected.

p3 l54-55: "for instance [...]" : this sentence could be reformulated to be more precise.

Thanks, it is corrected.

p4 l58: but also in the* torsional* direction

Thanks, it is corrected.

p4 l58: load is* applied

Thanks, it is corrected.

p4 l65: split the sentence: "in the figure. However, ..."

Thanks, it is corrected.

p4 l66: change sign through flapwise bending

Thanks, it is corrected.

p4 l66: alters blade loads and aseroelastic stability

Thanks, it is corrected.

p4 l73: It might be relevant to also cite the work of Wallrapp here, and put your work in perspective. Wallrapp also uses a Taylor expansion for the displacement field. For instance, the following references could be relevant: `https://doi.org/10.1002/nme.1620320818,https://link.springer.com/chapter/10.1007/978-94-017-0625-4_33,http://dx.doi.org/10.1080/08905459408905214`

Thanks for papers. Wallrapp's geometric stiffening paper is cited in 'relevant kinematics' section.

p4 l73: The term "Expansion modes" is not present in Table 1. Maybe it would make sense to introduce it in the table as well.

Thanks, it is added.

p5 l76: remove "through"

Thanks, it is corrected.

p6 eq 3: This eigenvalue problem assumes that the mass and stiffness matrices are constant (taken as the undeflected position). Maybe you could add some precision in the text. Would the method need to be modified if centrifugal stiffening effects are to be included? Would different shape functions be generated for different rotational speed?

It depends on user preference and application. On the other hand Flex or Fast doesn't use mode shapes with centrifugal stiffness effects since the rpm can be different for different load cases. However, they include centrifugal stiffening effects on the load vector by adding the bending loads due to centrifugal force and out of plane displacements. So, you can use the method in Flex or Fast with mode shapes computed at undeflected state.

See section-2

p6 l 97: Can you clarify what is meant by "tangential" in this sentence?

Thanks for the warning we have corrected the term as "tangent stiffness". The term "tangent" means $K_{tan} = \dfrac{dg(u)}{du}$ computed at $u = u_{current}$. There is also secant stiffness which is not our interest and it means $K_{sec} = \dfrac{g(u_{current})}{u_{current}}$ (no derivative) and for linear case it equals to tangent stiffness).

p7 l116: for convenience of communication → "for conciseness"

Thanks, it is corrected.

p7 l117: the derivative of the* stiffness matrix wrt the* jth

Thanks, it is corrected.

p7 l118: finite differences*, as* given in eq 8

Thanks, it is corrected.

p7 l120: is* computed

Thanks, it is corrected.

p7 l124: straightforward*

Thanks, it is corrected.

p7 l125: at the* undeflected

Thanks, it is corrected.

p7 l130: understanding of* the modal

Thanks, it is corrected.

p7 l135: the paragraph needs many grammatical corrections.

Thanks, it is corrected.

p7 l142: parenthesis are needed before the citation

Thanks, it is corrected.

p8 l144. In this study,* a*

Thanks, it is corrected.

p8 l146. In the* expansion mode

Thanks, it is corrected.

p8 eq10: the variable x is not introduced in the text and might not correspond to the one used in Figure 4&5.

Thanks, it is corrected.

p8 l 153: replace with: and the number of corresponding expansion modes ($M_{EM}$) is:

Thanks, it is corrected.

p9 l155-170. The procedure here is hard to follow. I would recommend rewriting it to make it clearer. - It is not clear to me why the amplitudes need to be computed based on a static formulation (with no inertial contributions). - I'm guessing you are describing a procedure to obtain Phi and $Phi_{EM}$ based on some "unit" static force fields. - You might want to address ealier in the text how Phi is determined in the expansion modes method (probably the same way as the modal derivative method, using eq 3). - How is $Phi_{EM}$ determined in equation 15? I'm understanding that for a given force field, u is obtained from a nonlinear code, and then q and $q_{em}$ are obtained by identification. Is that the case? But that means that Phi and $Phi_{EM}$ needs to be known somehow, no?

Thanks for your review. We updated section 3.2 according to your comments.

Please see section 3.2.

p9 l171: You mention that $Phi_{EM}$ can "also" be determined using least square, but it is not clear what this "also" means here. It was not clear in the lines above how $Phi_{EM}$ was determined. If there is indeed two ways to determine $Phi_{EM}$, you could use two separate subsections to distinguish the two methods. But most likely, equation 16 is actually the main method you are using to determine $Phi_{EM}$. As you can see, the reader can get quite confused, so I would advise to revise this section, trying to guide the reader more. For instance, stating earlier "The procedure to obtain $Phi_{EM}$ is as follows. First, we do X. Then Y".

Thanks for your comment. You are right that 'also' is confusing for defining the process. It is corrected.

p9 l174: Correct the sentence to: The modal expansion vectors are similar to the modal derivative vectors. They are identical (except for numerical differences) when ...

Thanks, it is corrected.

p9 l175: are more → is more

Thanks, it is corrected.

p9 l177: little→ small, clear → significant

Thanks, it is corrected.

p9 l178: On the other hand,* (comma)

Thanks, it is corrected.

p9 l179: this sentence should preferably be reformulated

Thanks, it is corrected.

p9 l181: as mentioned before, (comma)

Thanks, it is corrected.

p9 l181: this sentence is not clear. In general, you might want to expand this paragraph a little, to explain what is different in the quadratic correction vector method. You probably need to mention somewhere in the text (maybe early on), what is meant by "nonlinear stiffness terms", and "including quadratic vectors in the* reduction space". For instance, it was not clear to me why you mentioned "post-processing" on line 190. (the Phi* are computed as preprocessing steps, but the displacement field can indeed be computed as a postprocessing step).

Thanks for your review. The text is updated according to your reviews.

p10 l 190: shows* the* procedure to obtain the structural response with

Thanks, it is corrected.

p10 Algorithm: It seems that the Taylor expanded field does not introduce additional forces to the system because it only appears as a post processing step, which influences the displacement field (and thereby the external loads), but not the stiffness or inertia of the system. This is an important assumption that should be mentioned early on and discussed further.

Thanks for the review. 'Using constant stiffness and inertia matrix' is now mentioned in different places starting from introduction to method section.

p10: "Coupled response analysis (time-dependent)". "Time-dependent" is not clear here. In the "Pure structural response", the force was a function of time as well. I believe what is meant here is that the loads are nonlinear functions of q (or x)

Thanks for the correction. We deleted "(time-dependent)" text. The given formulation for "Pure structural response analysis" also show that q and F are also time dependent.

p11 l193: and the* IEA 15-MW

Thanks, it is corrected.

p11 l194-195: the sentence is not clear

Thanks, it is rephrased.

p15 eq 18: It is mentioned that "omega" is the rotation frequency of the beam. Is the beam rotating, or is it only the loads that are fluctuating with a given frequency? How are centrifugal stiffening effects accounted for if the beam is rotating?

Thanks, it is rephrased. So, there is no centrifugal forces but the $F_y$ computation is done by using the formula where $\omega$ is the rotational speed.

end of p15, and p21 l338: You observe that MD and EM have very similar results. My understanding is that these results be

identical in theory, no? The only difference here comes from the numerical method used to determine the first-order terms in the preprocessing step. I would recommend adding a small paragraph at the end of this result section to mention that (if I'm correct) or discuss it (if I'm wrong).

We thank the reviewer for the good question and suggestions. We confirm that there is similarity between MD and EM. This can be seen, for example, in Fig. 9. and Fig. 12. On the other hand, if the structure has coupled mode shapes due to geometry or material coupling, then MD and EM show small deviations from each other. Although the deviations are small, we believe they have numerical sources. The EM calculation includes many load cases with non-zero off diagonal terms in displacement and modal amplitude matrices for coupled mode shapes and they lead some deviation in the solution. MD calculation is performed for each modal derivative independently.

So an explanation is added after figure-12 about the source of the differences.

p18 l295: It seems the linear ROM predicts an elongation, whereas the other methods predict a compression (or the opposite). Could you discuss the reason for these differences here? (Also, and that is my misunderstanding, why is the linear ROM predicting a displacement in the z-direction? Is there a shape function in the elongation direction? It might be worth mentioning for other readers.)

We thank the reviewer for these questions. The Z-displacement is measured in beam root coordinate systems and each linear element keeps the tip position at same Z-position in their local frames. So, this gives non-zero z-displacement in root coordinate system for curved structures whose elements are not always aligned with root coordinate system. However, this is not observed for straight structures where the all elements are aligned with root coordinate system. In reality, IEA15 MW blade tip first goes in positive z-direction with flapwise bending force so the effective rotor diameter increases up to the point where the blade looks straight. After that point, blade tip z-position starts going into negative z-direction again. So there is still a chance to end up with positive z-displacement of tip compared to the initial z-position, but in this example blade is deflected by bending moments so much and the tip moved in negative z-direction compared to its initial position. You can also think in a way that, the initial positive z-displacement behaviour is captured by linear ROM but since there is no update by deflections, it estimates always positive z-displacement for tip by increasing bending deflection. So, the corrections are actually able to capture this complex behaviour change up to a point.

The related text is modified and extended to: "The linear ROM predicts a nonphysical elongation in the $z-$direction because of the non-zero motion in the $z-$direction in the linear modal shapes ( particularly Mode-1) shown in Fig. 11. In contrast, the two correction methods in this study are not limited by the linear modal shapes. Both of the correction methods provide a correct estimation of the shortening of the blade in the axial direction."

p21 l340: I think the results about 1.25m longer blade was not mentioned earlier in the results section (you might have used 1.2m), and I think this sentence is a bit unclear.

We thank the reviewer for this question. The exact difference between linear ROM and HAWC2 is mentioned as 1.29 meters.

In the revision, we have modified the sentence: "In other words, the linear ROM can predict a nonphysical elongation of the blade span." The rest sentences in this paragraph are also edited.